# Converting waste PET plastics into automobile fuels and antifreeze components

Zhiwen Gao[1,2], Bing Ma[1,2✉], Shuang Chen[1,2], Jingqing Tian[1,2] & Chen Zhao [1,2✉]

With the aim to solve the serious problem of white plastic pollution, we report herein a low-cost process to quantitatively convert polyethylene terephthalate (PET) into *p*-xylene (PX) and ethylene glycol (EG) over modified $Cu/SiO_2$ catalyst using methanol as both solvent and hydrogen donor. Kinetic and in-situ Fourier-transform infrared spectroscopy (FTIR) studies demonstrate that the degradation of PET into PX involves tandem PET methanolysis and dimethyl terephthalate (DMT) selective hydro-deoxygenation (HDO) steps with the in-situ produced $H_2$ from methanol decomposition at 210 °C. The overall high activities are attributed to the high $Cu^+/Cu^0$ ratio derived from the dense and granular copper silicate precursor, as formed by the induction of proper NaCl addition during the hydrothermal synthesis. This hydrogen-free one-pot approach allows to directly produce gasoline fuels and antifreeze components from waste poly-ester plastic, providing a feasible solution to the plastic problem in islands.

[1] Shanghai Key Laboratory of Green Chemistry and Chemical Processes, School of Chemistry and Molecular Engineering, East China Normal University, Shanghai 200062, China. [2] Institute of Eco-chongming, Shanghai 202162, China. ✉email: bma@chem.ecnu.edu.cn; czhao@chem.ecnu.edu.cn

Polyethylene terephthalate (PET) is the most abundant polyester plastic with nearly 70 million tons produced per year worldwide[1]. However, PET is not easily degradable after use[2,3]. Ten million tons of PET waste are discharged into the ocean annually, and the amount of PET residues from the textile and packaging industries is especially high[4,5]. As a result, the health of marine organisms is being seriously threatened[6,7]. Depolymerization and reuse of PET are the most common methods to solve this problem. Chemical depolymerization methods, mainly include hydrolysis, glycolysis, ammonolysis, and pyrolysis[8–11], can reserve the chemical composition of plastics and turn into stable monomer molecules. However, these methods still face limitations of harsh reaction conditions, low product yields, and purification difficulties.

Recent trends in solving marine plastic pollution have led to a proliferation of studies on PET chemical processing. Yan and Wang et al.[12] used $Ru/Nb_2O_5$ to convert PET to 87.1% yield of mixed arenes with 63% selectivity of p-xylene (PX) in water at 200 °C and 0.3 MPa $H_2$. Wang et al.[13] used the ethylene glycol fragments in the PET structure to provide a hydrogen source by aqueous-phase reforming, yielding 91.3% monomers with 19% selectivity of PX over $Ru/Nb_2O_5$ in water at 220 °C. However, the use of Ru increased the cost of the catalyst, and these catalytic systems led to an arene mixture product with a low selectivity to PX. Similarly, Yan et al.[14] demonstrated the conversion of PET into 70% yield of arenes in octane over a $Co/TiO_2$ catalyst at 340 °C and 3 MPa $H_2$. Li and Zhang et al.[15] converted PET to $C_7–C_8$ cycloalkanes and aromatics by a three-step reaction. Lately, photocatalytic and electrocatalytic reforming for PET waste upcycling becomes a novel strategy. Erwin et al.[16] reported the efficient photo-reforming of PET into $H_2$ and organic products such as formate, acetate, and pyruvate under ambient temperature. Duan et al.[17] reported electrocatalytic upcycling of PET over $CoNi_{0.25}P$ to potassium diformate, terephthalic acid, and $H_2$.

Chemical processing of other plastics is being also studied widely. Scott et al.[18] proposed a hydrogenolysis and aromatization method for converting waste polyethylene (PE) over a $Pt/\gamma$-$Al_2O_3$ catalyst into higher-value long-chain alkyl aromatics (80% yield) at 280 °C. Perras et al.[19] hydrogenolyzed high-density PE into a narrow distribution of diesel and lubricant-rang alkanes using an ordered mesoporous $Pt/SiO_2$ catalyst at 300 °C. The distribution of alkane products can be adjusted by modulating the sizes of the mesoporous pores of the catalysts.

However, in some islands especially with developed tourism, due to the lack of industry on the island, a large amount of abandoned plastic waste can only be disposed by landfill or incineration[20]. In this study, we developed a $H_2$-free one-pot method to directly convert PET and PBT wastes into gasoline fuels and antifreeze components using a low-cost Cu-based catalyst (Fig. 1). Through this process, methanol served as the solvent for methanolysis of PET to DMT and EG, and a source of hydrogen for DMT selective hydrodeoxygenation to PX. Upon addition of NaCl during the hydrothermal synthesis, the structure and formation process of $Cu/SiO_2$ catalyst were characterized. The reaction mechanism for overall PET conversion was studied through kinetic and in-situ Fourier-transform infrared spectroscopy (FTIR) simultaneously.

## Results and discussion
### Catalytic tests for hydrogen-free conversion of PET in alcohols.
The integrated system for PET depolymerization using alcohols as hydrogen donors contained several reaction steps: hydrogen generation by alcohol dehydrogenation as well as PET alcoholysis and monomer hydrodeoxygenation. Experiments demonstrated that the

alcoholysis of PET could be directly carried out in methanol at 210 °C in the absence of catalyst to obtain a 100% yield of DMT monomers within 30 min (Supplementary Fig. 1). Therefore, we mainly focused on catalyst development for methanol dehydrogenation and DMT hydrodeoxygenation for overall PET conversion in methanol. Table 1 lists the performances of different catalysts for converting of PET to downstream products in methanol at 210 °C for 6 h. Almost all the catalysts tested were inactive, except $Cu/SiO_2$ (HT) with a 73% yield of PX, while the yield of the by-product methyl 4-methylbenzoate and 4-methylbenzyl alcohol were 23% and 4%, respectively. Interestingly, $Co/SiO_2$, $Ni/SiO_2$, and $Fe/SiO_2$ showed no activity for the whole reaction. Once PET was depolymerized into DMT monomers, the hydrodeoxygenation reaction stopped since Co, Ni, and Fe active centers hardly catalyzed methanol dehydrogenation, resulting in a lack of hydrogen for DMT hydrodeoxygenation.

We subsequently used Cu as the metal active center to investigate the influence of the support on this reaction. Unlike $Cu/SiO_2$, $Cu/TiO_2$ (PX yield: 17%), $Cu/CeO_2$ and $Cu/ZrO_2$ (PX yield: 0%) did not show good performances. We speculated that $SiO_2$ had a more amorphous structure than $TiO_2$, $CeO_2$, and $ZrO_2$ and was easily etched by ammonia to form a strong copper silicate structure. This copper silicate precursor structure was beneficial to partially reduce to $Cu^+$ and $Cu^0$ for methanol dehydrogenation.

The synthesis methods of $Cu/SiO_2$ were investigated by comparing the effects of various Cu-based catalysts on the conversion of PET. Several Cu-based catalysts were prepared using various methods such as the hydrothermal method (HT), impregnation method (IM), deposition–precipitation with urea (DPU), and deposition–precipitation with ammonia (DPA). Only the $Cu/SiO_2$ catalysts prepared by HT and DPA showed reactivity towards methanol dehydrogenation, producing hydrogen at 2.9 and 0.8 MPa, respectively. As indicated above, this hydrogen can be used for subsequent DMT hydrodeoxygenation. However, hydrodeoxygenation of PET on $Cu/SiO_2$ (DPA) stopped at intermediate methyl 4-methylbenzoate, most likely because of the lack of sufficient $H_2$ released during the methanol dehydrogenation to completely convert PET to PX.

Unlike the HT and DPA methods, the Cu species of the catalyst prepared by the IM and DPU methods generated $Cu^0$ species after being completely reduced, as shown by XRD patterns (Supplementary Fig. 2). In addition, Cu X-ray photoelectron spectroscopy (XPS) (Supplementary Fig. 3a) indicated that $Cu^{2+}$ was incompletely reduced on HT and DPA samples. The Cu LMM X-ray induced Auger spectra (XAES) (Supplementary Fig. 3b) showed that the ratios of $Cu^+/Cu^0$ in DPU, DPA and IM samples were significantly lower than that ratio in HT sample. In previous work[21–23], it was proven that a mixture of combined $Cu_2O$ and Cu was the active site for methanol dehydrogenation. This could explain the low activity of $Cu/SiO_2$ (IM) and $Cu/SiO_2$ (DPU) to provide the required $H_2$ for DMT hydrodeoxygenation reactions. In a next step, we attempted to introduce alkali metals (e.g., Na, Li, K, Rb, and Cs) in the form of chlorides via a hydrothermal treatment to investigate the catalyst activity. A 100% PX yield was obtained on $CuNa/SiO_2$ (HT) (Supplementary Fig. 4). Thus, the addition of NaCl significantly promoted the conversion of PET to PX on $Cu/SiO_2$ (HT).

Later, the temperature effects for alcoholysis of PET and further hydrodeoxygenation of PET with $Cu/SiO_2$ (HT) at 170–210 °C were displayed in Supplementary Tables 1 and 2. At 200 °C, PET was completely depolymerized to DMT and EG in methanol, but PX yield was lowered to 93.6%. It is also stated in the supporting notes that the methanol consumption calculated from the product well matches the actual consumption of methanol and the in-situ generated hydrogen.

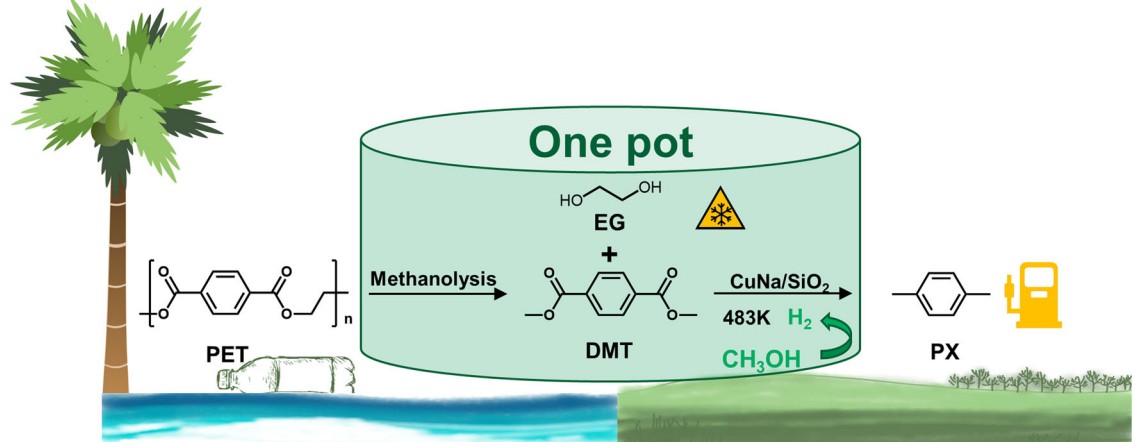

**Fig. 1 Strategy.** Strategy for the depolymerization and conversion of PET wastes.

**Table 1 Catalytic performance of different catalysts used in the PET conversion process.**

| Catalyst | PET Conv. (%) | DMT Yield (%) | PX yield (%) | By-product yield (%) | | Incremental pressure at RT (MPa) | Gas composition (%) | | | |
|---|---|---|---|---|---|---|---|---|---|---|
| | | | | Methyl 4-methylbenzoate | 4-methylbenzyl alcohol | | H₂ | CO | CH₄ | CO₂ |
| Cu/SiO₂ (HT) | 100 | – | 73 | 23 | 4.0 | 2.9 | 58 | 39 | 3 | |
| Co/SiO₂ | 100 | 100 | – | – | – | – | – | – | – | – |
| Ni/SiO₂ | 100 | 100 | – | – | – | – | – | – | – | – |
| Fe/SiO₂ | 100 | 100 | – | – | – | – | – | – | – | – |
| Cu/TiO₂ | 100 | – | 16 | 57 | 27 | 1.2 | 69 | 28 | 2 | 1 |
| Cu/ZrO₂ | 100 | 100 | – | – | – | – | – | – | – | – |
| Cu/CeO₂ | 100 | 100 | – | – | – | – | – | – | – | – |
| Cu/SiO₂ (IM) | 100 | 100 | – | – | – | – | – | – | – | – |
| Cu/SiO₂ (DPU) | 100 | 100 | – | – | – | – | – | – | – | – |
| Cu/SiO₂ (DPA) | 100 | 97.8 | – | 2.2 | – | 0.8 | 53 | 35 | 12 | – |
| CuNa/SiO₂ | 100 | – | 100 | – | – | 3.4 | 60 | 36 | 4 | – |
| CuLi/SiO₂ | 100 | – | 89 | 8.4 | 2.2 | 3.0 | 62 | 35 | 3 | – |
| CuK/SiO₂ | 100 | – | 97 | 1.9 | 0.7 | 3.3 | 58 | 38 | 4 | – |
| CuRb/SiO₂ | 100 | – | 60 | 18 | 22 | 3.0 | 63 | 34 | 3 | – |
| CuCs/SiO₂ | 100 | – | 67 | 20 | 13 | 3.0 | 62 | 33 | 5 | – |

Reaction conditions: 0.12 g PET, 0.1 g catalyst, 30 mL methanol, 210 °C, 6 h.
*HT* hydrothermal method, *IM* impregnation method, *DPA* deposition–precipitation with ammonia method, *DPU* deposition–precipitation with urea method, *RT* room temperature, *EG* ethylene glycol, *DMT* dimethyl terephthalate, *PX* p-xylene.

In the subsequent experiments, ethanol and isopropanol were tested as solvents and hydrogen donors for converting of PET as well (Supplementary Table 3). Experimental data showed that PET can be well alcoholyzed at 210 °C in both solvents in absence of catalysts, obtaining 80.3% and 73.5% yields of monomers after 0.5 h, respectively. However, in the further PET catalytic conversion tests over CuNa/SiO₂ (HT), the gained monomers from PET were not hydrogenated and no p-xylene was formed in ethanol or isopropanol solvents, probably due to the lack of sufficient released hydrogen in this catalytic system. To extend the plastic availability, we also tested another plastic polybutylene terephthalate (PBT) under the same system and the gained results were very similar to PET conversion (Supplementary Table 4). At 210 °C, 100% yields of p-xylene and 1,4-butanediol were obtained from PBT in methanol, releasing 2.8 MPa gases (60% H₂).

**Structural characterization of CuNa/SiO₂ catalyst.** To explore the high activities of PET conversion on the CuNa/SiO₂ catalyst, a series of characterization was conducted. Cu/SiO₂ (dried) and CuNa/SiO₂ (dried) samples prepared by the HT method exhibited X-ray diffraction (XRD) peaks characteristics of Cu₂Si₂O₅(OH)₂ (2θ = 19.9, 21.8, 30.8, 35.0, 57.5, and 62.4°) (PDF #27-0188)

(Fig. 2a)[24,25]. The addition of NaCl reduced the crystallinity of copper silicate and the size of Cu nanoparticles, resulting in more uniform size distribution. The peaks for copper silicate disappeared after air-calcination and hydrogen-reduction, and they were replaced by Cu characteristics peaks (2θ = 43.3° (PDF #04-0836)) and Cu₂O (2θ = 36.4, 42.3, 61.3, and 77.3° (PDF #05-0667)) (Supplementary Fig. 5). The N₂ adsorption–desorption showed that the specific surface area and mesoporous volume of the Cu/SiO₂ (dried) precursor were 277.9 m² g⁻¹ and 0.08 cm³ g⁻¹, respectively (Fig. 2b). After the addition of Na⁺, the specific surface area decreased by 5/6 (46.9 m² g⁻¹), and the mesoporous volume also decreased significantly (0.06 cm³ g⁻¹). Overall, the entire structure became denser.

The hydrogen temperature-programmed reduction (H₂-TPR) profile of the dried Cu/SiO₂ precursor sample showed reduction peaks at 256 and 280 °C (Supplementary Fig. 6a), ascribed to the reduction of copper silicate to Cu₂O·SiO₂ and Cu⁰, respectively. After the addition of NaCl, reduction peaks appeared at higher temperatures of 274 and 299 °C, revealing a stronger interaction between Cu particles and the support. H₂-TPR revealed that this dense structure was relatively difficult to reduce. The decreased CO adsorption on FTIR spectra at 2125 and 2111 cm⁻¹ on the

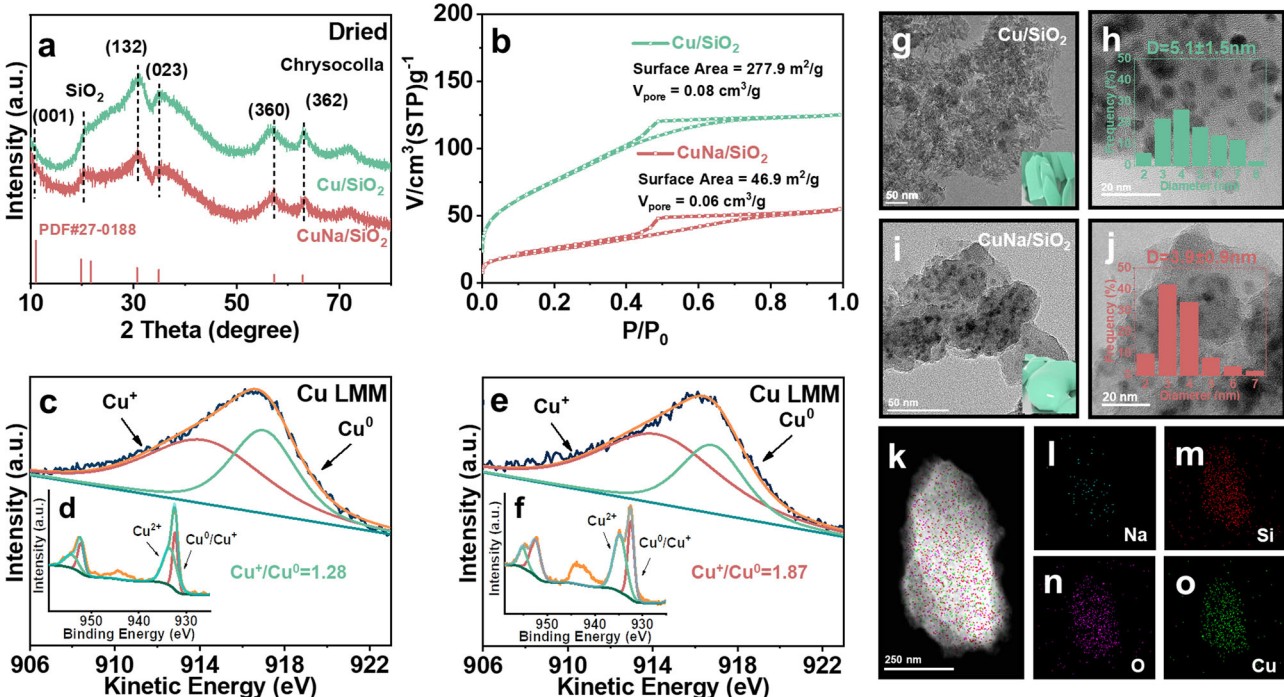

**Fig. 2 Structural characterizations. a** X-ray diffraction (XRD) patterns of Cu/SiO$_2$ (dried) and CuNa/SiO$_2$ (dried). **b** N$_2$ adsorption–desorption of Cu/SiO$_2$ (dried) and CuNa/SiO$_2$ (dried). X-ray photoelectron spectroscopy (XPS) Cu 2$p$ spectra of **d** Cu/SiO$_2$ (reduced) and **f** CuNa/SiO$_2$ (reduced). Cu LMM X-ray induced Auger electron spectroscopy (XAES) spectra of **c** Cu/SiO$_2$ (reduced) and **e** CuNa/SiO$_2$ (reduced). **g** Transmission electron microscopy (TEM) images of Cu/SiO$_2$ (dried), **h** Cu/SiO$_2$ (reduced) at a higher magnification, **i** CuNa/SiO$_2$ (dried), and **j** CuNa/SiO$_2$ (reduced) at higher magnification. **k** Transmission electron spectroscopy–energy dispersive X-ray spectroscopy (TEM-EDS) mappings of the elements in CuNa/SiO$_2$: (**l**) Na, (**m**) Si, (**n**) O, and (**o**) Cu.

Cu$^+$ sites of reduced CuNa/SiO$_2$ compared to Cu/SiO$_2$ (Supplementary Fig. 6b), probably due to the highly decreased surface area of the CuNa/SiO$_2$ sample.

We measured the FTIR spectra of different states of Cu/SiO$_2$ and CuNa/SiO$_2$ (Supplementary Fig. 6c–e) and found that both dried samples showed a characteristic O-H stretching vibration peak at 669 cm$^{-1}$, which was ascribed to the copper silicate species. These results are in line with the XRD results. The peak at 795 cm$^{-1}$ was attributed to the bending vibrations of the Si-O bond of the amorphous silica support. The relative content of copper silicate was determined by the intensities of two peaks (i.e., I$_{669}$/I$_{795}$). After air-calcination, the intensity of the characteristic peak of copper silicate decreased slightly, which may be explained in terms of a lower crystallinity since copper silicate lost part of the crystal water during the calcination process. Moreover, in the reduced sample the characteristic peak at 669 cm$^{-1}$ nearly disappeared, revealing that majority of copper silicate may have been reduced to Cu species by hydrogen.

The thermogravimetric analysis (TGA) profile of the precursor (Supplementary Fig. 7) showed that physisorbed water from the copper silicate precursor was removed at a temperature lower than 130 °C. As the temperature increased to 600 °C, the crystal water was gradually removed, and the copper silicate decomposed into CuO and SiO$_2$. After the addition of NaCl, the water content of the copper silicate precursor was decreased by 6.09%, implying that the copper silicate structure became more compact.

The surface compositions of reduced samples were calculated from Cu XPS and Auger Cu LMM spectra[26,27], as showed in Supplementary Table 5. The total surface contents of Cu (T(Cu)) as obtained from XPS analysis on Cu/SiO$_2$ and CuNa/SiO$_2$ were 4.68% and 5.92%, respectively. These values were smaller than those as tested by ICP-AES (Supplementary Table 6), since XPS detected the surface Cu species while ICP measured the total Cu contents. Cu$^{2+}$ satellite peaks at 940–950 eV of XPS revealed an incomplete reduction of the copper silicate precursor (Fig. 2d and f). The ratio of Cu$^{2+}$/T(Cu) in CuNa/SiO$_2$ (0.44%) was much larger than Cu/SiO$_2$ (0.37%), due to its higher difficulty in reduction of copper silicate (Supplementary Table 5). Since the Cu$^0$ and Cu$^+$ species from Cu 2$p_{3/2}$ (932.1 eV) and Cu 2$p_{1/2}$ (952.2 eV) are too close to distinguish[27,28], we intuitively determined the Cu$^+$/Cu$^0$ ratio by Cu LMM X-ray induced Auger electron spectroscopy (XAES, Fig. 2c and e). The higher Cu$^+$/Cu$^0$ ratio (1.87) of CuNa/SiO$_2$ confirmed that after the addition of Na$^+$, copper silicate with a dense texture was less likely to be reduced to Cu$^0$. A higher ratio of Cu$^+$/Cu$^0$ was indicative of a higher activity tendency to both methanol dehydrogenation and DMT hydrodeoxygenation[21–23].

Transmission electron microscopy (TEM) images intuitively showed the different morphologies of the two copper silicates formed with and without NaCl introduction during the hydrothermal synthesis process. Thus, while the dried precursor of Cu/SiO$_2$ showed a layered copper silicate structure (Fig. 2g), the dried precursor of CuNa/SiO$_2$ showed a special state of granular particle accumulation (Fig. 2i). After reduction under H$_2$, a high-resolution transmission electron microscopy (HRTEM) revealed a Cu particle size distribution in CuNa/SiO$_2$ centered at 3.9 ± 0.9 nm (Fig. 2j), while Cu/SiO$_2$ showed smaller Cu particle sizes (5.1 ± 1.5 nm) (Fig. 2h). TEM-mapping confirmed that Cu and Na were uniformly distributed on SiO$_2$ (Fig. 2o and l). SEM (Supplementary Fig. 8) showed that CuNa/SiO$_2$ had more small granular particles compared to Cu/SiO$_2$ after air calcination and hydrogen reduction.

Compared with traditional Cu/SiO$_2$, the granular copper silicate with poor crystallinity formed after the addition of Na$^+$ had a very dense structure, with a specific surface area of 46.9 m$^2$ g$^{-1}$ and a mesopore volume of 0.06 cm$^3$ g$^{-1}$. It is proposed that the addition of NaCl inhibits nucleation of layered copper silicate and normally grown of this phase into a complete crystal shape, and finally exhibiting a state of granular particle accumulation.

**Influence of $Na^+/Cu^{2+}$ molar ratio towards the formed CuNa/$SiO_2$ catalysts and corresponding activities**. To further explore the influence of the addition of NaCl on the formation of the catalyst, we carried out a series of characterization tests. $Na^+/Cu^{2+}$ Molar ratios of 2.5:1, 5:1, 10:1, and 15:1 were denoted as 2.5 NaCl, 5 NaCl, 10 NaCl, and 15 NaCl, respectively. The precursor samples all showed the characteristic diffraction peaks of the $Cu_2Si_2O_5(OH)_2$ crystal phase (Supplementary Fig. 9), indicating that the addition of NaCl did not affect the phase composition of the catalyst. In the case of the 5 NaCl sample, the copper silicate showed poor crystallinity compared to other samples (Supplementary Table 7). $N_2$ adsorption–desorption (Supplementary Fig. 10) revealed that CuNa/$SiO_2$ had the lowest surface area (46.9 $m^2$ $g^{-1}$) upon addition of 5 NaCl (Supplementary Fig. 11a), indicating that the formed structure was the most compact among the samples tested herein. TGA tests of the CuNa/$SiO_2$ precursor showed that physisorbed water (2.41%) and crystal water (6.75%) upon addition of 5 NaCl were the lowest among all the samples tested (Supplementary Fig. 12 and Supplementary Fig. 11b). This also confirmed that the copper silicate structure was densest at this ratio. In the XPS ($Cu2p$) and Cu LMM XAES profiles of the CuNa/$SiO_2$ after reduction, the ratio of $Cu^+/Cu^0$ still presented a volcano-type distribution (Supplementary Fig. 11c and Supplementary Table 8), and attaining the highest $Cu^+/Cu^0$ ratio of CuNa/$SiO_2$ (1.86) when 5 NaCl was introduced (Supplementary Fig. 13).

In a subsequent step, we investigated the influence of the different $Na^+/Cu^{2+}$ molar ratio generated during the hydrothermal treatment on the performance of CuNa/$SiO_2$. Based on the activity tests on PET at 210 °C, the yield of PX exhibited a maximum value at 100% on 5 NaCl sample, while samples of 2.5 NaCl, 10 NaCl, and 15 NaCl showed lower yields of 78.3, 92.3, and 60.7%, respectively (Supplementary Fig. 11d).

Finally, we tried to use the optimal Cu-5Na/$SiO_2$ for cyclic reaction tests (Supplementary Table 9). The used catalyst can still maintain a PX yield of 96.4% from PET conversion in the second bath, but when the third bath was performed, the PX yield was reduced to 52.7%. The catalyst deactivation was probably due to the enlarged Cu particle size after recycling tests as verified by XRD patterns (Supplementary Fig. 14) and TEM images (Supplementary Fig. 14b, c), and the reduction of Cu species (Supplementary Fig. 14d, e) by the excess hydrogen produced via methanol dehydrogenation. XAES analysis proved that the ratio of $Cu^+/Cu^0$ drastically decreased from 1.87 to 0.57 after four runs (Supplementary Fig. 14f), which hindered the synergistic effect of $Cu^0$ and $Cu^+$ for the catalytic process.

**Structural formation mechanism on CuNa/$SiO_2$ catalysts**. In the traditional hydrothermal synthesis process (Fig. 3a), $Cu^{2+}$ in the solution combined with the silanol on the $SiO_2$ surface to form copper silicate, which accelerated the layered copper silicate nucleation and growth significantly. This type of layered copper silicate leads to a low $Cu^+/Cu^0$ ratio on the layered copper silicate surface (Fig. 2c).

Upon addition of 5 NaCl during hydrothermal synthesis (Fig. 3b), some amounts of $Na^+$ occupied the silanol on the surface of the $SiO_2$, thereby inhibiting nucleation and growth of layered copper silicate (Fig. 2g, i). $Cu^{2+}$ in the solution could only be combined with the remaining silanol on the $SiO_2$ surface to form scattered and isolated copper silicate particles, and the compact structure had a small surface area and poor crystallinity. The formed granular copper silicate showed a large interface area with $SiO_2$, since no remaining Si-OH on CuNa/$SiO_2$ was detected by IR spectra in vacuum (Supplementary Fig. 15). Granular copper silicate was more difficult to reduce compared with traditional layered copper silicate, resulting in a high ratio of $Cu^+/Cu^0$ active sites in the prepared catalyst (Supplementary Fig. 11c).

However, when the amount of added 15 NaCl was too high during the hydrothermal synthesis (Fig. 3c), $Na^+$ occupied all the silanol sites on $SiO_2$, resulting in the precipitation of $Cu^{2+}$ with $SiO_3^{2-}$ in solution to form copper silicate, which was then deposited on the $SiO_2$ surface. Compared to the catalyst with 5 NaCl introduced during hydrothermal treatment, this type of copper silicate showed better crystallinity (Supplementary Table 7) and small interface areas with $SiO_2$ (Supplementary Fig. 15), and was relatively easier to be reduced to $Cu/Cu_2O·SiO_2$, forming low ratio of $Cu^+/Cu^0$ (Supplementary Fig. 11c).

To ascertain whether the addition of NaCl only affected the formation process of copper silicate or could promote the reaction itself, we prepared a Cu/$SiO_2$-HT-Na-IM sample. We first synthesized Cu/$SiO_2$ by a hydrothermal method, which was subsequently impregnated with NaCl after the formation of layered copper silicate. The new copper silicate precursor had the same loading of $Na^+$ (2.4%) (Supplementary Table 10) as CuNa/$SiO_2$. XRD (Supplementary Fig. 16) showed that the impregnated $Na^+$ did not affect the formation of copper silicate, although the yield of PX was moderate (65.8%). Thus, we confirmed that the addition of NaCl in the hydrothermal treatment affected the morphology of the copper silicate and therefore the $Cu^+/Cu^0$ ratio after reduction. Na impregnation after the formation of copper silicate not only failed to promote the catalyst activity but also covered the active sites, resulting in a reduction of PX yield.

In general, the addition of 5 NaCl in the hydrothermal treatment resulted in the formation of granular copper silicate with a lower crystallinity, smaller specific surface area, and denser texture. When the molar mass ratio of $Na^+/Cu^{2+}$ reached 5:1, the $Cu^+/Cu^0$ ratio reached the maximum (1.86), providing significantly more active sites for methanol dehydrogenation and DMT hydrodeoxygenation.

**The reaction path of DMT hydro-deoxygenation to PX in methanol**. We used the optimal Cu-5Na/$SiO_2$ to conduct a kinetics study on the reaction of DMT (A) and the intermediates at the optimal reaction temperature (210 °C) and monitored the distribution of products over time. As soon as the reaction started, the DMT concentration decreased (Fig. 4a) at an initial rate of 0.36 g $g^{-1}$ $h^{-1}$, revealing a high efficiency for hydrogen production. Hydrogen was produced during the heating process, and it was sufficient to maintain the amount of hydrogen required for the subsequent reaction. When the reaction started, the intermediate methyl 4-methylbenzoate (C) was produced and a maximum yield of 24.3% at 1.5 h. Within 1–1.5 h, the intermediate 4-methylbenzyl alcohol (D) was produced slowly. At 3 h, almost all the DMT was converted, while the yield of the target product PX reached 100% at 6 h.

In this kinetics study, we only observed two intermediates: C and D. Importantly, we did not observe 1,4-benzenedimethanol, which implied that DMT underwent one-sided adsorption on CuNa/$SiO_2$. Therefore, methyl 4-(methylol)benzoate (B) may appear transitorily as an intermediate product. Based on these results, we speculated that the reaction from DMT to PX involved four steps: (1) one-sided adsorption of the ester of DMT on CuNa/$SiO_2$ and subsequent hydrogenation to alcohol, yielding B; (2) alcohol of B underwent hydrogenolysis to methyl groups and desorbed to form C; (3) ester C adsorbed on CuNa/$SiO_2$ and was hydrogenated to alcohol and obtain D; (4) alcohol D underwent hydrogenolysis to methyl groups and desorbed to obtain the target product PX. The kinetics of the three intermediates were studied under the same conditions (Fig. 4b–d). Intermediates A, B, C, and D were completely consumed after approximately 5, 1.5,

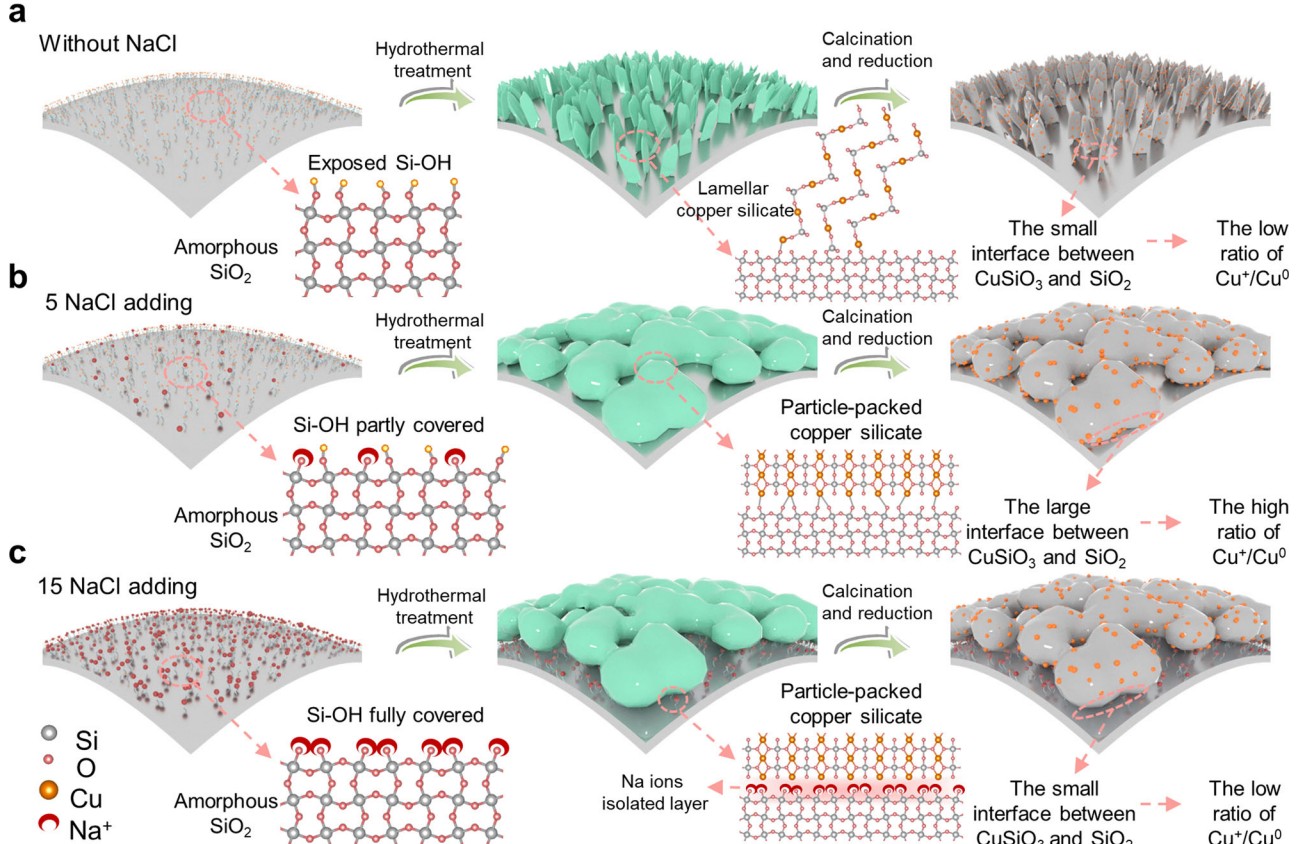

**Fig. 3 Structural formation processes. a** Without NaCl adding, **b** with 5 NaCl adding, **c** with 15 NaCl adding copper silicate formation processes with different amounts of introduced NaCl.

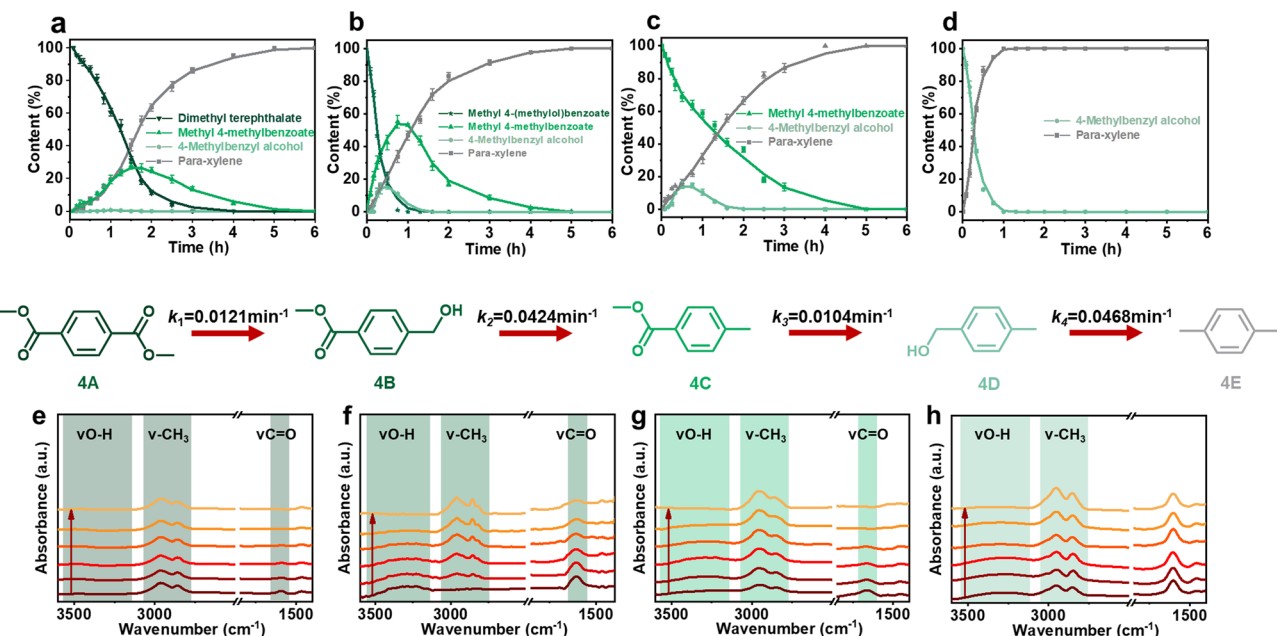

**Fig. 4 Kinetic and in-situ Fourier-transform infrared spectroscopy (FTIR) studies.** Product distribution–reaction time curves for the catalytic conversion of: **a** dimethyl terephthalate (A), **b** methyl 4-(methylol)benzoate (B), **c** methyl 4-methylbenzoate (C), and **d** 4-methylbenzyl alcohol (D) on CuNa/SiO₂. Reaction conditions: PET, 0.12 g; CuNa/SiO₂ catalyst, 0.1 g; methanol, 30 mL; 210 °C; 6 h. Data are presented as mean ± s.d. of three independent experiments. Time-resolved in-situ transmitted Fourier-transform infrared spectroscopy difference spectra of **e** dimethyl terephthalate (A), **f** methyl 4-(methylol)benzoate (B), **g** methyl 4-methylbenzoate (C), and **h** 4-methylbenzyl alcohol (D) conversion CuNa/SiO₂ in methanol over at 120 °C from 10 to 60 min.

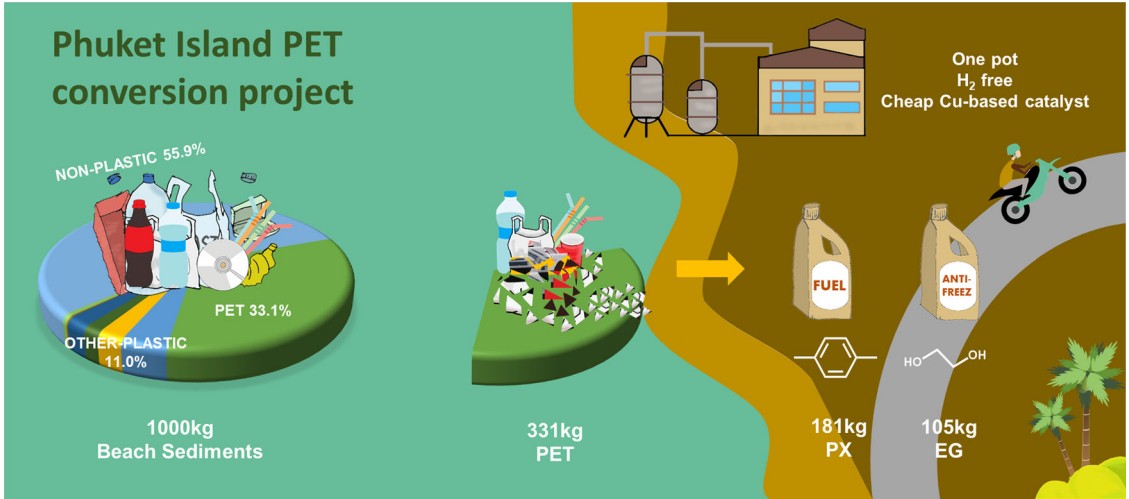

**Fig. 5 Preliminary on-site test.** Schematic diagram for application of new PET conversion project on sediments from Phuket Island.

5, and 1 h, respectively. The experimental results confirmed our proposed reaction pathway. The simulation results performed using MATLAB yielded the rate constants of each step ($k_1 = 0.0121 \, \text{min}^{-1}$, $k_2 = 0.0424 \, \text{min}^{-1}$, $k_3 = 0.0104 \, \text{min}^{-1}$, $k_4 = 0.0468 \, \text{min}^{-1}$) (Supplementary code 1). It is worth noting that C was obtained with the largest concentration after the DMT hydrogenation. This was because steps (2) and (3) required intermediate C desorption and re-adsorption on CuNa/SiO$_2$, making the hydrogenation of C the rate-determining step ($k_3 = 0.0104 \, \text{min}^{-1}$) of the overall process. The rate of alcohol hydrogenolysis was about 4 times that of the ester hydrogenation.

In-situ FTIR also demonstrated our reaction process for converting of DMT and intermediates (B, C, and D) on CuNa/SiO$_2$. The different substrate functional groups appeared in four regions in the FTIR spectra: (1) aryl C=C, 1494, 1523 cm$^{-1}$, (2) aryl C=O, 1594–1664 cm$^{-1}$, (3) aryl -CH$_3$ stretching, 2950–2863 cm$^{-1}$, and (4) O-H, 3305–3290 cm$^{-1}$. It should be noted that the aryl C=C band of the four substrates at 1494 and 1523 cm$^{-1}$ did not change during the reaction (Fig. 4e–h), revealing that the aromatic structure remained intact during the process, and aromatic hydrocarbons tended to be generated. The intensity of the C=O stretching vibration peak at 1594–1664 cm$^{-1}$ for intermediates A, B, and C decreased continuously with time until complete vanishment (Fig. 4e–g). This phenomenon indicated that the hydrogenation reaction occurred continuously under in-situ conditions. A and C, which lacking hydroxyl groups itself, produced O-H vibration peaks at 3305–3290 cm$^{-1}$ and then gradually disappeared (Fig. 4e and g), indicating that C=O hydrogenation to hydroxyl groups occurred, then hydrogenolysis took place, in line with the kinetics results. In addition, the bands corresponding to hydroxyl groups of B and C gradually decreased until they vanished as a result of hydrogenolysis. Finally, the -CH$_3$ vibration peak of PX continuously increased with time. The in-situ infrared study once again confirmed the path of DMT conversion on CuNa/SiO$_2$, and the results are highly consistent with the kinetics behavior.

**Calculation of environmental energy impact for PET conversion.** Based on the data described above, we tried to compare the efficiency of PET conversion over CuNa/SiO$_2$ in this work with the parallel literature using the similar methodology. Thielemans et al.[29] introduced a useful method to quantifiably assess the different depolymerization conditions for PET according to three green chemistry metrics, which are energy economy ($\varepsilon$ coefficient), the environmental factor (E) and the environmental energy impact ($\xi$).

The best processes would tend to present high $\varepsilon$ coefficient and low values of $E_{factor}$ and $\xi$. Supplementary Table 11 clearly showed that this work using base metal catalysts had the highest $\varepsilon$ (1.323E-5 °C$^{-1}$*min$^{-1}$) due to its high product yield (100%), low reaction temperature (210 °C) and time (360 min) compared to the similar PET conversion processes over Ru/Nb$_2$O$_5$ catalysts. We also tried to redouble the PET and catalyst amounts at the same time, and still attained 100% PX yield, leading to the lowest $E_{factor}$ (13.41) and $\xi$ (1013605 °C*min). Such comparison demonstrates that the current implementation for chemical depolymerization of PET is highly efficient.

**On-site test for PET recycling.** Finally, we conducted a preliminary on-site test of an island using our PET transformation method (Fig. 5). A recent survey of beach sediment along the coastline of the Phuket Island showed that PET (mainly containing beverage bottles, plastic films, and microwave packaging) accounted for ca. 33.1% of the overall plastic sediment (Supplementary Fig. 17)[30]. Several common PET plastics that are available on the tourist island were chosen to convert, such as Coca-Cola bottles, McDonald's drink caps, disposable lunch boxes, packaging bags and even some polyester clothes (Supplementary Fig. 18a). After simple treatment with the raw materials with scissors (Supplementary Fig. 18b), we obtained 100% yield of p-xylene from different sources of PET plastics at the same catalytic conditions. Every ton of plastic sediment contained 331 kg of PET, and thus, 181 kg of PX and 105 kg of ethylene glycol (EG) could be obtained via this route under optimal conditions. These products and methanol can be easily separated by simple distillation (Supplementary Table 12). The obtained PX and EG could be used as automobile fuel and antifreeze replenishment.

In this work, we demonstrate herein that low-cost CuNa/SiO$_2$ provides a viable option for processing waste PET and PBT accumulated on islands without a need of external hydrogen, transforming it into a high-value-added energy supply. The system integrates in situ hydrogen production from methanol dehydrogenation as well as PET methanolysis and subsequent DMT hydrodeoxygenation to PX. Such multi-function is realized on a Cu/SiO$_2$ catalyst with a high Cu$^+$/Cu$^0$ ratio derived from reduction of the dense and granular copper silicate precursor. This developed green chemical recycling process on poly-ester plastic could be applied on islands with scarce resources and high accumulation of ocean plastics, which directly provides vehicle energy supply and thus benefits the economy of the islands.

## Methods

**Catalyst preparations.** Synthesis of CuNa/SiO$_2$ using hydrothermal method. Cu(NO$_3$)$_2$ · 3H$_2$O was dissolved in deionized water. NaCl and SiO$_2$ were added to the fully dissolved solution, which was stirred at room temperature for 0.5 h and ultrasonically treated for 0.5 h. The mixture was then transferred to a polytetra-fluoroethylene autoclave and kept at 120 °C in a homogeneous reactor for 3 h. After cooling to room temperature, the solid in the autoclave was filtered and washed with deionized water to pH = 7. The obtained precursor was dried overnight, calcined at 450 °C in an air atmosphere for 4 h (air flow rate: 150 mL·min$^{-1}$, heating rate: 2 °C·min$^{-1}$), and then reduced at 450 °C in a hydrogen atmosphere for 4 h (hydrogen flow rate: 150 mL·min$^{-1}$, heating rate: 2 °C·min$^{-1}$).

**Activity tests.** The typical experiment was carried out as follows: 0.12 g PET, 0.1 g catalyst, and 30 mL methanol were loaded into a batch autoclave (60 mL). The autoclave was purged with N$_2$ to remove the residual air at ambient temperature. The reaction was conducted at 210 °C for 6 h at a stirring speed of 600 rpm. The liquid products were analyzed by a Shimadzu GC coupled with GC-MS equipped with a Rtx-5Sil MS capillary column (30 m × 0.25 mm × 0.25 μm). The gas products were analyzed by a Shimadzu GC-2014 chromatographic automatic sample injection analysis. A 60 cm long TDX-01 packed column with an inner diameter of 2 mm was used to separate the components, and the TCD was used as a detector.

## Data availability

All relevant data supporting the key findings of this study are available within the Supplementary Information files and Source Data or from the corresponding author upon request. Source data are provided with this paper.

## Code availability

The software is provided as supplementary code with the article file.

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

## Acknowledgements

This research was supported by the National Key Research and Development Program of China (Grant No. 2016YFB0701100) and the Recruitment Program of Global Young Experts in China.

## Author contributions

Z.G.: Experiments perform, Formal analysis, Writing - original draft. B.M.: Formal analysis, Graphic design, writing. S.C.: Formal analysis. J.T.: Conceptualization, Formal analysis. C.Z.: Conceptualization, Formal analysis, Funding acquisition, Resources, Supervision, Writing - original draft.

## Competing interests

The authors declare no competing interests.
