## [Peer Review File · Nature Communications]

Converting waste PET plastics into automobile fuels and antifreeze componentsReviewers' Comments:

Reviewer #1:

Remarks to the Author:

The article describes hydrogen gas-free conversion of PET plastic to para-xylene and glycol. In itself the work is interesting, but many questions remain regarding the catalytic compounds, rationale, comparison with competing technology, as well as experimental details that I would like the authors to address. I have listed them below.

1. The rationale for converting PET waste into fuel baffles me a bit. In this work PET is converted first to glycol and DMT, after which DMT is converted to pX for fuel applications. This would mean that valuable resources, especially aromatics, will just be burnt rather than reused. Given the limited amount of resources available on the planet one should focus on reusing resources rather than burning them. Please discuss the rationale of the work in light of this in the introduction.
2. PET depolymerization has recently been reviewed by Barnard, Rubio and Thielemans in Green Chemistry. This work also introduces a methodology to quantifiably compare different processes based on materials use (using Sheldon's factor) and energy consumption. Please compare the efficiency of this work with the existing literature using this (or a similar) methodology.
3. PET recycling in the introduction is heavily focused on conversion to fuel, whereas depolymerization into monomers would economically make more sense (less steps and no loss of resources) yet is not covered. Please add this to the introduction.
4. Values are reported without error bars based on repeat experiments. Please add error bars to experimental results from repeat experiments to allow the reader to assess significance in differences.
5. XRD results are interpreted qualitatively which does not allow for objective and quantifiable comparison. Please carry out Rietveld refinement or another method to allow for a numerical comparison between results.
6. XPS analysis: Cu⁺ and Cu⁰ peaks are indeed convoluted. To deconvolute, ratio's between peaks should be kept constant in Auger, as well as the 952.2 and 932.1 eV peaks, and be cross-correlated with the Cu₂O contribution in the O 1s signal. Please make sure this is indeed the case and describe the methodology in SI.
7. Line 99: How was partial reduction of Cu⁺ to Cu⁰ measured? How much is partial reduction? Please quantify.
8. Line 112: XRD data in Figure 2S do not say anything about surface species, only bulk species. Please provide and describe surface analysis info as additional info.
9. Line 122: what is the proof for the Cu reduction?
10. Line 167: The way this is written seems like Cu⁺ has a peak at 952.2 eV and Cu⁰ at 932.1 eV. Please rewrite and also specify the orbital assignment.
11. Line 197-198: "smaller and more uniformly distributed". It's very subjective to conclude this from SEM images. Different people can/will see different things.
12. Line 203: what is a volcano-type distribution? It seems just like a trend going through a maximum...
13. Line 210: "poor crystallinity". Please quantify (Rietveld refinement...)
14. Line 225: "low interface area" How was this determined? Quantification?
15. Line 227: "inhibiting nucleation and growth" How was this determined? Please provide proof.
16. Line 231: "large interface area" How was this determined? Quantification?
17. What is special about the 5:1 ratio Na⁺/Cu²⁺ that this would give rise to the optimal Cu⁺/Cu⁰ ratio?
18. Please provide full analysis of the Phuket plastic waste sample.
19. SI chemicals: Please add chemical purities
20. SI catalyst preparation: washing has been done until pH is 7 but ions can still leach then. Until what eluent conductivity was washing carried out?
21. SI line 67: Pore assumptions for BJH calculations?
22. SI line 70: TEM conditions?
23. SI line 99: XPS needs more information: e.g. charge neutralization (and method)?, Sensitivity factor?,...

24. SI line 105: ICP calibration method used?

25. Figure S2: Please add the initial HT sample result as well for comparison.

Reviewer #2:

Remarks to the Author:

PET contributes significantly to plastic waste generation. In this work, Zhao and co-workers developed a new H₂ free method using Cu based catalyst to convert PET into xylene and ethylene glycol. The conversion efficiency is high and the reaction pathway has been well studied. The catalyst structure-activity correlation has also been convincingly described. Overall it is a nice piece of study that deserves to be published in Nat. Commun. after proper revision.

1) The stability of the catalyst. I feel this is a major limitation of the work. There is no adequate information on the stability of the Cu catalyst. Detailed characterizations of the spent Cu catalyst may be provided. Further, the reusability of the catalyst may be studied in more detail. If direct reuse is not possible, what is the reason for catalyst deactivation and whether it is possible to recover catalyst activity by certain treatment.

2) The authors may also comment on the applicability of the catalytic system beyond PET. If it is only applicable to PET, then sorting strategies have to be applied.

3) Methanol is used as solvent and hydrogen donor. How much methanol is decomposed during the process? Does the consumption of methanol match that of PET conversion?

4) It would also be good if the authors comment on how to purify products and reuse unreacted methanol.

5) It is interesting to see the example of Phuket Island. Did the authors really use samples collected there, or it is just based on literature report? This needs to be made clear. If the sample from Phuket Island is not used, then it should be removed from Figure 5. Related parts in MS should be modified as well.

Reviewer #3:

Remarks to the Author:

The manuscript "Converting waste PET plastics into automobile fuel and antifreeze components" by Zhao and coworkers describes a novel methodology for the depolymerization of PET waste using methanol as the solvent and hydrogen source, and a Cu-based catalyst. This new method is very important due to the use of an alcohol as reducing agent and also a non-toxic and earth abundant metal catalyst.

The work is well written and the discussion of the results appropriate, involving an extensive study of reaction conditions. Some aspects of the reaction mechanism were also included in this study.

I recommend the publication of this manuscript in nature communications, after major revisions:

-Authors should test this method using other alcohols, for example ethanol and isopropanol, as the hydrogen source and verify that PET depolymerization and p-xylene formation also occur.

- Authors should also indicate whether the PET alcoholysis reaction can be carried out at temperatures below 210 °C and what yields are obtained.

- To study the applicability of this method, it would be very interesting if the authors tested this method from the second most used polyester, polybutylene terephthalate (PBT), and checked if it is also possible to obtain p-xylene with good yields.

2022.04.01.

We are grateful for the constructive comments from three reviewers, and based on these suggestions, we have modified and improved our manuscript. The detailed responses to the comments of the reviewers are listed in blue together with the text of the original material.

Manuscript ID: NCOMMS-21-40636A

Title: Converting waste PET plastics into automobile fuel and antifreeze components

Reviewer #1:

The article describes hydrogen gas-free conversion of PET plastic to para-xylene and glycol. In itself the work is interesting, but many questions remain regarding the catalytic compounds, rationale, comparison with competing technology, as well as experimental details that I would like the authors to address. I have listed them below.

1. The rationale for converting PET waste into fuel baffles me a bit. In this work PET is converted first to glycol and DMT, after which DMT is converted to PX for fuel applications. This would mean that valuable resources, especially aromatics, will just be burnt rather than reused. Given the limited amount of resources available on the planet one should focus on reusing resources rather than burning them. Please discuss the rationale of the work in light of this in the introduction.

Reply: Thanks for the Reviewer's suggestion. We agree with you that plastic waste is one of the most valuable wastes and can be considered as a potentially cheap source for the production of industrial fuels and chemicals. In land cities, it is feasible to converted plastics into chemical raw materials through effective chemical

recycling methods. However, in some islands, especially those with developed tourism, due to the lack of industry on the island, a large amount of abandoned plastic waste can only be disposed by landfill or incineration. Recycled plastics are also generally limited to feasible applications where low-quality materials are collected, resulting in minimal economic incentives for waste recycling, sorting and processing. At this time, if it can be used as a raw material to output gasoline energy and antifreeze components on the island through a simple process, it will be a very practical way.

2. PET depolymerization has recently been reviewed by Barnard, Rubio and Thielemans in Green Chemistry. This work also introduces a methodology to quantifiably compare different processes based on materials use (using Sheldon's factor) and energy consumption. Please compare the efficiency of this work with the existing literature using this (or a similar) methodology.

Reply: Thanks for the Reviewer's suggestion. We tried to use the environmental factor and environmental energy impact in the Barnard, Rubio and Thielemans's work to evaluate the efficiency of several parallel works. Firstly, energy economy coefficient (ϵ) is proposed to enable objective comparison on the influence of parameters such as temperature, catalyst type, or proportion of starting materials, where t is the reaction time (in minutes), T is the reaction temperature in degrees celsius, and Y is the yield of the main monomer in mass fraction (which containing the aromatic moiety) in eqn (1). Barnard et al. improved the environmental factor (E_{factor}) in eqn (4) which took the effect of materials input that results in waste generation into consideration. The environmental energy impact factor (ξ) results from the combination of the two factors above as presented in eqn (5). The best processes would tend to present low values of E_{factor} and ξ factor and high ϵ values.

Table R1 clearly showed that this work has the highest ϵ ($1.323\text{E-}5^{\circ}\text{C}^{-1}\cdot\text{min}^{-1}$) due to its excellent product yield (100%) and low reaction temperature (210 °C) and time (360 min). High solvent/PET ratio (197.5) resulted in the high E_{factor} (37.19). But ξ ($2811035^{\circ}\text{C}\cdot\text{min}$) is still the smallest by combining the above two coefficients, and about twice time than other works. We also tried to redouble the PET and catalyst at the same time, and still get 100% PX yield, greatly reduced E_{factor} as well as ξ .

In addition, only this work uses non-noble metal catalysts, and the obtained products are highly selective.

$$\varepsilon = \frac{Y}{T \times t} \quad (1)$$

$$E_{\text{factor}} = \frac{[0.1 \times (\frac{\text{solvent}}{\text{PET}} \text{ ratio}) + (\frac{\text{cat}}{\text{PET}} \text{ ratio}) + (\text{other} \frac{\text{subst}}{\text{PET}} \text{ ratio})] \times m_{\text{PET}}}{m_{\text{Product}}} \quad (2)$$

$$m_{\text{Product}} = \text{yield}_{\text{Product}} \times \frac{MM_{\text{Product}}}{MM_{\text{PET}} \text{ MERE}} \times m_{\text{PET}} \quad (3)$$

Replacing (3) in (2)

$$E_{\text{factor}} = \frac{[0.1 \times (\frac{\text{solvent}}{\text{PET}} \text{ ratio}) + (\frac{\text{cat}}{\text{PET}} \text{ ratio}) + (\text{other} \frac{\text{subst}}{\text{PET}} \text{ ratio})] \times m_{\text{PET}}}{\text{yield}_{\text{Product}} \times \frac{MM_{\text{Product}}}{MM_{\text{PET}} \text{ MERE}} \times m_{\text{PET}}} \quad (4)$$

$$\xi = \frac{E_{\text{factor}}}{\varepsilon} \quad (5)$$

Table R1. Summary of main results in literature for PET conversion.

Catalyst	Noble metal	T (°C)	Reaction time (min)	Yield arene (%)	Solvent/PET mass ratio	Catalyst/PET mass ratio	Energy economy (ε) (°C ⁻¹ *min ⁻¹)	Environmental factor (a.u)	Environmental energy impact (ξ) (°C*min)	Ref.
Ru/Nb ₂ O ₅	Yes	220	720	91.3%	75.0	1.00	5.764E-6	19.91	3454198	1
Ru/Nb ₂ O ₅	Yes	280	480	83.6%	133.3	1.00	6.220E-6	34.20	5498392	2

CuNa/SiO ₂	No	210	360	100%	197.5	0.83	1.323E-5	37.19	2811035	This work
CuNa/SiO ₂	No	210	360	100%	98.75	0.83	1.323E-5	19.40	1466364	This work
CuNa/SiO ₂	No	210	360	100%	65.83	0.83	1.323E-5	13.41	1013605	This work

Ref.

[1] Lu, S. *et al.* H₂-free Plastic Conversion: Converting PET back to BTX by Unlocking Hidden Hydrogen. *ChemSusChem* **14**, 4242-4250 (2021).

[2] Jing, Y. *et al.* Towards the Circular Economy: Converting Aromatic Plastic Waste Back to Arenes over a Ru/Nb₂O₅ Catalyst. *Angewandte Chemie International Edition* **60**, 5527-5535, (2021).

3. PET recycling in the introduction is heavily focused on conversion to fuel, whereas depolymerization into monomers would economically make more sense (less steps and no loss of resources) yet is not covered. Please add this to the introduction.

Reply: Thanks for the Reviewer's suggestion. We have added the depolymerization of PET into monomers in introduction.

Chemical depolymerization methods, mainly include hydrolysis, glycolysis, and ammonolysis^[3-6], can reverse the chemical composition of plastics and turn into stable monomer molecules again. However, these methods still face limitations of harsh reaction conditions, low product yield, and purification difficulties.

Ref.

[3] Ügdüler, S. *et al.* Towards closed-loop recycling of multilayer and colored PET plastic waste by alkaline hydrolysis. *Green Chemistry* **22**, 5376-5394 (2020).

[4] Kang, M. J., Yu, H. J., Jegal, J., Kim, H. S. & Cha, H. G. Depolymerization of PET into terephthalic acid in neutral media catalyzed by the ZSM-5 acidic catalyst. *Chemical Engineering Journal* **398**, 125655 (2020).

[5]Mori, H. *et al.* Organosoluble Oligomer Obtained by Glycolysis of Poly(ethylene terephthalate) and Its Detailed Structural Characterization by MALDI-TOF Mass Spectrometry. *Polymer Journal* **34**, 687-691 (2002).

[6]Mittal, A., Soni, R. K., Dutt, K. & Singh, S. Scanning electron microscopic study of hazardous waste flakes of polyethylene terephthalate (PET) by aminolysis and ammonolysis. *Journal of Hazardous Materials* **178**, 390-396 (2010).

4. Values are reported without error bars based on repeat experiments. Please add error bars to experimental results from repeat experiments to allow the reader to assess significance in differences.

Reply: Thanks for the Reviewer's suggestion. We repeated each experiment three times, and the error bars are added in following figures.

Supplementary Fig. 9 (a) Effect of different contents of introduced NaCl on *p*-xylene (PX) yield.

Figure 4. Product distributions–reaction time curves for the catalytic conversion of (a) dimethyl terephthalate, (b) methyl 4-(methylol)benzoate, (c) methyl 4-methylbenzoate, (d) 4-methylbenzyl alcohol on CuNa/SiO₂.

5. XRD results are interpreted qualitatively which does not allow for objective and quantifiable comparison. Please carry out Rietveld refinement or another method to allow for a numerical comparison between results.

Reply: Thanks for the Reviewer’s suggestion. Due to the small particle size, poor crystallinity and weak intensity, XRD results maybe not suitable for Rietveld refinement. We compared the results with the standard spectrum (PDF#27-0188) in the database [7-8], and the results are shown in Figure 2a.

Figure 2. (a) XRD patterns of Cu/SiO₂ (dried) and CuNa/SiO₂ (dried).

Ref.

[7] Yue, H. *et al.* A copper-phyllsilicate core-sheath nanoreactor for carbon–oxygen hydrogenolysis reactions. *Nature Communications* **4**, 2339 (2013).

[8] Wang, Z.-Q. *et al.* High-Performance and Long-Lived Cu/SiO₂ Nanocatalyst for CO₂ Hydrogenation. *ACS Catalysis* **5**, 4255-4259 (2015).

6. XPS analysis: Cu⁺ and Cu⁰ peaks are indeed convoluted. To deconvolute, ratio's between peaks should be kept constant in Auger, as well as the 952.2 and 932.1 eV peaks, and be cross-correlated with the Cu₂O contribution in the O 1s signal. Please make sure this is indeed the case and describe the methodology in SI.

Reply: Thanks for the Reviewer's suggestion. Concerning on the analysis of distribution of Cu⁺ and Cu⁰ peaks in catalysts, we have rewritten the corresponding sentences as follows:

“Since the Cu⁰ and Cu⁺ species from the Cu 2p_{3/2} (932.8eV) and Cu 2p_{1/2} (952.7 eV) are too close to well distinguish, we intuitively determined the Cu⁺/Cu⁰ ratio

by Cu LMM X-ray induced Auger electron spectroscopy (XAES, Figure 2c and e). The relative analysis of Cu^+ and Cu^0 ratios in XAES, please refer to question 7.”

Figure R1. XPS patterns of O 1s of the reduced CuNa/-SiO₂ (reduced) and Cu/SiO₂ (reduced).

The O1s XPS of reduced catalysts was conducted, as shown in Figure R1. The peak at 530.3 eV was observed in the reduced CuNa/SiO₂ and Cu/SiO₂, ascribed to Cu₂O or CuSiO₃ as reported.^[9-10] The peak of 532.5 eV was attributed to the O 1s of SiO₂ support^[10]. However, we cannot distinguish the Cu₂O and CuSiO₃ species through the O1s XPS analysis.

Ref.

[9] Ding, J. *et al.* Highly selective and stable Cu/SiO₂ catalysts prepared with a green method for hydrogenation of diethyl oxalate into ethylene glycol. *Applied Catalysis B: Environmental* **209**, 530-542 (2017).

[10] Elzey, S. *et al.* Formation of paratacamite nanomaterials via the conversion of aged and oxidized copper nanoparticles in hydrochloric acidic media. *Journal of Materials Chemistry*, **21**, 3162-3169 (2011).

[11] Zhang, Z. *et al.* Ti³⁺ Tuning the Ratio of Cu⁺/Cu⁰ in the Ultrafine Cu Nanoparticles for Boosting the Hydrogenation Reaction. *Small* **17**, 2008052, (2021).

7. Line 99: How was partial reduction of Cu⁺ to Cu⁰ measured? How much is partial reduction? Please quantify.

Reply: Thanks for the Reviewer's suggestion. The compositions of reduced catalysts were calculated from XPS and XAES analysis [12-13], as shown in Supplementary Table 4. The total surface contents of Cu (T(Cu)) as obtained from XPS analysis on Cu/SiO₂ and CuNa/SiO₂ were 4.68% and 5.92%, respectively. These values were smaller than those as tested by ICP-AES, since XPS detected the surface Cu species while ICP measured the total Cu contents. Cu²⁺ satellite peaks at 940–950 eV of XPS revealed an incomplete reduction of the precursor (Figures 2d and f). The ratio of Cu²⁺/T(Cu) (0.44%) in CuNa/SiO₂ was much larger than Cu/SiO₂ (0.37%), due to the higher difficulty in reduction. The X-ray induced Auger spectra (XAES) Cu LMM were employed to distinguish between the Cu⁰ and Cu⁺ and the deconvolution results are listed. The higher Cu⁺/Cu⁰ ratio (1.87) of CuNa/SiO₂ confirmed that after the addition of Na⁺, copper silicate with a low crystallinity and a dense texture was less likely to be reduced to Cu⁰. A higher ratio of Cu⁺/Cu⁰ was indicative of a higher tendency to both methanol dehydrogenation and DMT hydrodeoxygenation. This part has been modified in the revised version.

Supplementary Table 4. Mass coverage (wt%) on the catalyst surfaces resulting from XPS (Cu2p) peak and Cu LMM XAES peak deconvolution.

Catalyst	XPS			XAES		
	T(Cu) ^a	Cu ²⁺ ^a	$\frac{Cu^{2+}}{T(Cu)}$ ^a	Cu ⁺ ^b	Cu ⁰ ^b	$\frac{Cu^+}{Cu^++Cu^0}$ ^b
Cu/SiO ₂	4.68	1.73	0.37	1.66	1.29	0.56
CuNa/SiO ₂	5.92	2.62	0.44	2.15	1.15	0.65

Note: ^a T(Cu) is the total surface content of Cu, obtained from XPS characterization.

^b Intensity ratio between Cu⁺ and (Cu⁺+ Cu⁰) by deconvolution of Cu LMM XAES spectra.

Ref.

[12] Wen, C. *et al.* Reaction temperature controlled selective hydrogenation of dimethyl oxalate to methyl glycolate and ethylene glycol over copper-hydroxyapatite catalysts. *Applied Catalysis B: Environmental* **162**, 483-493 (2015).

[13] Ding, J. *et al.* Highly selective and stable Cu/SiO₂ catalysts prepared with a green method for hydrogenation of diethyl oxalate into ethylene glycol. *Applied Catalysis B: Environmental* **209**, 530-542 (2017).

At the same time, we also analyzed different CuNa/SiO₂ samples by the same methods, as shown in Supplementary Table 7.

Supplementary Table 7. Mass coverage (wt%) on the catalyst surfaces resulting from XPS (Cu2p) peak and Cu LMM XAES peak deconvolution.

Catalyst	XPS			XAES	
	T(Cu) ^a	Cu2+ ^a	Cu+ ^b	Cu0 ^b	$\frac{Cu^+}{Cu^++Cu^0}$ ^b
Cu2.5Na/SiO ₂	4.38	2.46	0.95	0.97	0.49
Cu5Na/SiO ₂	5.92	2.62	2.15	1.15	0.65
Cu10Na/SiO ₂	4.61	2.31	1.26	1.04	0.55
Cu15Na/SiO ₂	4.38	1.59	1.38	1.41	0.49

Note: ^a T(Cu) is the total surface content of Cu, obtained from XPS characterization.

^b Intensity ratio between Cu+ and (Cu⁺+ Cu0) by deconvolution of Cu LMM XAES spectra.

8. Line 112: XRD data in Figure 2S do not say anything about surface species, only bulk species. Please provide and describe surface analysis info as additional info.

Reply: Thanks for the Reviewer's suggestion. To investigate the surface information of Cu/SiO₂ samples prepared by different methods, we used XPS and Auger Cu

LMM analysis to analyze the distributions of copper species on their surfaces (Supplementary Fig. 3). On the XPS profiles of HT and DPA samples, there are obvious Cu^{2+} satellite peaks (940-950 eV) (Supplementary Fig. 3a), indicating that Cu^{2+} was incompletely reduced. However, the same phenomenon does not appear in the XPS of DPU and IM Cu/SiO_2 samples. The Cu LMM X-ray induced Auger spectra (XAES) were employed to distinguish between the Cu^0 and Cu^+ species (Supplementary Fig. 3b). The results showed that the ratios of Cu^+/Cu^0 in DPU, DPA and IM samples were significantly lower than such ratio in the HT sample, which corresponds to a significant decrease in reactivity of methanol dehydrogenation and PET hydrodeoxygenation (Table 1). This part has been revised in the renewed version.

Supplementary Fig. 3 (a) XPS and (b) Cu LMM XAES of Cu/SiO_2 (reduced) prepared by hydrothermal method (HT), deposition-precipitation with ammonia (DPA), deposition-precipitation with urea (DPU), and an impregnation method (IM).

9. Line 122: what is the proof for the Cu reduction?

Reply: Thanks for the Reviewer's suggestion. The XRD patterns of CuNa/SiO_2 after three times of reaction runs (Supplementary Fig. 16) can prove that lots of the Cu^+ species is reduced to Cu^0 . We have revised this part.

Supplementary Fig. 16 (a) XRD patterns of used-catalyst after 4 times and fresh CuNa/SiO₂ (reduced).

10. Line 167: The way this is written seems like Cu⁺ has a peak at 952.2 eV and Cu⁰ at 932.1 eV. Please rewrite and also specify the orbital assignment.

Reply: Thanks for the Reviewer's suggestion. We have already fixed the typographical error and rewrite it as below.

New version: Since the Cu⁰ and Cu⁺ species from the Cu 2p_{3/2} (932.1 eV) and Cu 2p_{1/2} (952.2 eV) are too close to distinguish the peaks^[14-15], we intuitively determined the Cu⁺/Cu⁰ ratio by Cu LMM X-ray induced Auger electron spectroscopy (XAES, Figures 2c and 2e).

Ref.

[14] Wen, C. *et al.* Reaction temperature controlled selective hydrogenation of dimethyl oxalate to methyl glycolate and ethylene glycol over copper-hydroxyapatite catalysts. *Applied Catalysis B: Environmental* **162**, 483-493 (2015).

[15] Sun, J. *et al.* Freezing copper as a noble metal-like catalyst for preliminary hydrogenation. *Science Advances* **4**, eaau3275 (2018).

11. Line 197-198: “smaller and more uniformly distributed”. It’s very subjective to conclude this from SEM images. Different people can/will see different things.

Reply: Thanks for the Reviewer’s suggestion. We have revised it as “SEM (Supplementary Fig. 7) showed that CuNa/SiO₂ had more small granular particles compared to Cu/SiO₂.”.

12. Line 203: what is a volcano-type distribution? It seems just like a trend going through a maximum...

Reply: Thanks for the Reviewer’s suggestion. We have rewritten this part. Based on the conversion tests of PET at 210 °C, the yield of PX exhibited a maximum PX yield at 100% for the 5 NaCl sample, while samples 2.5 NaCl, 10 NaCl, and 15 NaCl showed lower yields of 78.3, 92.3, and 60.7%, respectively (Supplementary Fig. 9a).

13. Line 210: “poor crystallinity”. Please quantify

Reply: Thanks for the Reviewer’s suggestion. We used the software MDI-Jade to fit the XRD data of samples firstly, and then calculated the crystallinity of each sample. The crystallinity and R-values (fitting error) are listed below.

Supplementary Table 6. The crystallinity of different Cu/SiO₂ catalysts.

Catalyst	Crystallinity (%)	R (%)
Cu/SiO ₂	45.1	1.51
Cu2.5Na/SiO ₂	45.0	1.56
Cu5Na/SiO ₂	23.6	1.17
Cu10Na/SiO ₂	27.0	1.40
Cu15Na/SiO ₂	27.1	1.31

14 and 15. Line 225: “low interface area” How was this determined? Quantification? Please provide proof.

Line 231: “large interface area” How was this determined? Quantification?

Reply: In order to roughly quantify the interface areas of the formed copper silicate with SiO₂, we supplemented the measurement of IR spectroscopy in vacuum and determined the remaining silanols groups on SiO₂ (Supplementary Fig. 14). The peak at 3740 cm⁻¹ in the SiO₂ sample is attributed to Si-OH groups (Supplementary Fig. 14) [16-17]. With the traditional hydrothermal method, Cu²⁺ in the solution combined with the silanol on the SiO₂ surface to form copper silicate, which accelerated the layered copper silicate nucleation and growth significantly. Thus only a small amount of Si-OH can still be detected. Upon addition of 5 NaCl, a large amount of Na⁺ occupied the silanol on the surface of the SiO₂, Cu²⁺ in the solution could only be combined with the remaining silanol on the SiO₂ surface to form scattered and isolated copper silicate particles (Figures 2g and 2i), which was attached to the surface of the carrier. Thus, the formed granular copper silicate showed a large interface area with SiO₂, since no remaining Si-OH on CuNa/SiO₂ was detected by IR spectra in vacuum (Supplementary Fig. 14). When 15 Na⁺ was introduced, Na⁺ occupied almost all the Si-OH on the surface, and Cu²⁺ can only combined with SiO₃²⁻ in the solution to form granular copper silicate and then deposited on the SiO₂. After washed with deionized water, some Si-OH groups are exposed, the peak at 3740 cm⁻¹ is reserved. Therefore, upon adding 15 Na⁺, this type of copper silicate showed

better crystallinity (Supplementary Table 6) and small interface areas with SiO₂ (Supplementary Fig. 14).

This part has been modified accordingly.

Supplementary Fig. 14 IR spectra in a vacuum.

Ref.

[16] Nakamura, M., Kobayashi, M., Kuzuya, N., Komatsu, T. & Mochizuka, T. Hydrophilic property of SiO₂/TiO₂ double layer films. *Thin Solid Films* **502**, 121-124 (2006).

[17] Tada, H. Layer-by-layer construction of SiO_x film on oxide semiconductors. *Langmuir* **11**, 3281-3284 (1995).

16. Line 227: “inhibiting nucleation and growth” How was this determined?

Reply: Thanks for the Reviewer's suggestion. Transmission electron microscopy (TEM) images intuitively showed the different morphologies of the two copper silicates formed with and without NaCl introduction during the hydrothermal process. Thus, while the dried precursor of Cu/SiO₂ showed a layered copper silicate structure (Figure 2g), the dried precursor of CuNa/SiO₂ showed a special state of granular particle accumulation (Figure 2i). Therefore, it is inferred that Na⁺ introduction indeed inhibits the nucleation and growth of layered copper silicate precursor.

Figure 2. Transmission electron microscopy (TEM) images of (g) Cu/SiO₂ (dried), (i) CuNa/SiO₂ (dried).

17. What is special about the 5:1 ratio Na⁺/Cu²⁺ that this would give rise to the optimal Cu⁺/Cu⁰ ratio?

Reply: Thanks for the Reviewer's suggestion.

When the Na⁺/Cu²⁺ ratio is 5:1, the obtained granular copper silicate has the smallest specific surface area, the lowest water content and the poorest crystallinity (Supplementary Fig. 10-13). TGA tests of the CuNa/SiO₂ precursor showed that physisorbed water (2.41%) and crystal water (6.75%) upon addition of 5 NaCl was the lowest among all the samples tested (Supplementary Fig. 11). This also confirmed that the copper silicate structure was densest at this ratio. N₂ adsorption-desorption

(Supplementary Fig. 12) revealed that CuNa/SiO₂ had the lowest surface area (46.9 m²/g) upon addition of 5 NaCl, indicating that the formed structure was the most compact among the samples tested herein.

A large amount of Na⁺ occupied the silanol on the surface of the SiO₂ upon addition of 5 NaCl, thereby inhibiting nucleation and growth of layered copper silicate (Figures 2g and 2i). Cu²⁺ in the solution could only be combined with the remaining silanol on the SiO₂ surface to form scattered and isolated copper silicate particles, and the compact structure had a small surface area and poor crystallinity. This granular copper silicate grown from the surface of SiO₂ is more stable and difficult to be reduced, resulting in a higher Cu⁺/Cu⁰ ratio and benefiting the further methanol dehydrogenation and DMT hydrodeoxygenation.

However, when the amount of added NaCl was too high, Na⁺ occupied all the silanol sites on SiO₂, resulting in the precipitation of Cu²⁺ with SiO₃²⁻ in solution to form copper silicate, which was then deposited on the SiO₂ surface. Compared to the catalyst with 5 NaCl introduced during hydrothermal treatment, this type of copper silicate showed better crystallinity (Supplementary Table 6) and was relatively easier to be reduced to Cu/Cu₂O·SiO₂ with a low ratio of Cu⁺/Cu⁰ (Supplementary Fig. 9d). In general, the addition of NaCl in the hydrothermal treatment resulted in the formation of granular copper silicate with a lower crystallinity, smaller specific surface area, and denser texture, which leads to form an optimal Cu⁺/Cu⁰ ratio after reduction.

Supplementary Fig. 10 X-ray diffraction (XRD) patterns of dried samples: (a) Cu-2.5Na/SiO₂, (b) Cu-5Na/SiO₂, (c) Cu-10Na/SiO₂, and (d) Cu-15Na/SiO₂.

Supplementary Fig. 11 TGA patterns of dried samples: (a) Cu-2.5Na/SiO₂, (b) Cu-5Na/SiO₂, (c) Cu-10Na/SiO₂, and (d) Cu-15Na/SiO₂.

Supplementary Fig. 12 N₂ adsorption-desorption of dried samples: (a) Cu-2.5Na/SiO₂, (b) Cu-5Na/SiO₂, (c) Cu-10Na/SiO₂, and (d) Cu-15Na/SiO₂.

Supplementary Fig. 13 Cu LMM XAES spectra of reduced samples: (a) Cu-2.5Na/SiO₂, (b) Cu-5Na/SiO₂, (c) Cu-10Na/SiO₂, and (d) Cu-15Na/SiO₂.

18. Please provide full analysis of the Phuket plastic waste sample.

Reply: Thanks for the Reviewer's suggestion. A recent survey of beach sediment along the coastline of the Phuket Island showed that PET (mainly containing beverage bottles, plastic films, and microwave packaging) accounted for ca. 33.1% of the overall plastic sediment (Supplementary Fig. 17). Several common PET plastics that are available on the tourist island were chosen to convert, such as Coca-Cola bottles, McDonald's drink caps, disposable lunch boxes, packaging bags and even some polyester clothes (Supplementary Fig. 18a). After simple treatment with the raw materials by scissors (Supplementary Fig. 18b), we obtained 100% yield of *p*-xylene from different sources of PET plastics at the same catalytic conditions.

Considering that lots of plastic wastes on the island sediments landfill are mixed together, in this context, we used CuNa/SiO₂ to catalyze the mixtures of PET and another plastic PBT at 210 °C, and the results showed that mixed plastics can be completely converted to *p*-xylene as well.

Supplementary Fig. 17 Compositions in sediments along the coast of Phuket island.

Figure5. Results of the application of our new PET conversion catalytic route on sediment from the Phuket Island.

Supplementary Fig. 18 (a) Disposable lunch boxes, Coca-Cola bottles, packaging bags, polyester clothes and McDonald's drink caps; (b) Fragments of the plastics after processing; (c) The snapshots of the reaction system.

19. SI chemicals: Please add chemical purities.

Reply: Thanks for the Reviewer's suggestion. We added all the chemical purities.

New version: The following chemicals were received from a commercial supplier and directly used without any pretreatment: nano silicon dioxide (SiO_2 , $\geq 99.9\%$, Sinopharm Chemical Reagent Co., Ltd.), ammonium chloride (NH_4Cl , $\geq 99.8\%$, Sinopharm Chemical Reagent Co., Ltd.), sodium chloride (NaCl , $\geq 99.8\%$, Sinopharm Chemical Reagent Co., Ltd.), polyethylene terephthalate (PET, Sinopharm Chemical Reagent Co., Ltd.), copper nitrate ($\text{Cu}(\text{NO}_3)_2 \cdot 3\text{H}_2\text{O}$, 99.0~102.0%, Sinopharm Chemical Reagent Co., Ltd.), ammonia ($\text{NH}_3 \cdot \text{H}_2\text{O}$, AR, Sigma-Aldrich), lithium chloride ($\text{LiCl} \cdot \text{H}_2\text{O}$, $\geq 97.0\%$, Sinopharm Chemical Reagent Co., Ltd.), chlorinated potassium (KCl, 99.8%, Aladdin), rubidium chloride (RbCl , 99.95%, Aladdin), cesium chloride (CsCl , 99.99%, Aladdin), cobalt nitrate ($\text{Co}(\text{NO}_3)_2 \cdot 3\text{H}_2\text{O}$, 99%, Aladdin), nickel nitrate ($\text{Ni}(\text{NO}_3)_2 \cdot 3\text{H}_2\text{O}$, $\geq 98.0\%$, Sinopharm Chemical Reagent Co., Ltd.), iron nitrate ($\text{Fe}(\text{NO}_3)_3 \cdot 9\text{H}_2\text{O}$, $\geq 98.5\%$, Sinopharm Chemical Reagent Co., Ltd.), dimethyl terephthalate ($\text{C}_{10}\text{H}_{10}\text{O}_4$, 99%, Sigma-Aldrich), methyl 4-(hydroxymethyl)benzoate ($\text{C}_9\text{H}_{10}\text{O}_3$, 98%, Accela), methyl *p*-toluate ($\text{C}_9\text{H}_{10}\text{O}_2$, 99%, InnoCHEM), 4-methylbenzyl alcohol ($\text{C}_8\text{H}_{10}\text{O}$, $\geq 99\%$, InnoCHEM), titanium dioxide (TiO_2 , $\geq 98.0\%$, Sinopharm Chemical Reagent Co., Ltd.), zirconium dioxide (ZrO_2 , $\geq 99.0\%$, Sinopharm Chemical Reagent Co., Ltd.), and cerium dioxide (CeO_2 , $\geq 99.9\%$, Sinopharm Chemical Reagent Co., Ltd.).

20. SI catalyst preparation: washing has been done until pH is 7 but ions can still leach then. Until what eluent conductivity was washing carried out?

Reply: Thanks for the Reviewer's suggestion. Until pH=7, we added a small amount of deionized water, and measure the conductivity of the liquid after filtration was $4.2 \pm 0.3 \mu\text{S}/\text{cm}$. This part has been supplemented into the supporting information part.

21. SI line 67: Pore assumptions for BJH calculations?

Reply: Thanks for the Reviewer's suggestion. The BJH method assumes that the thickness of adsorbent layer in the capillary is dependent only on the relative pressure P/P_0 , but not on the adsorbent properties and pore radius.

22. SI line 70: TEM conditions?

Reply: Thanks for the Reviewer's suggestion. Concerning on the TEM conditions, the samples prepared by grinding and subsequent dispersing the powder in ethanol and applying a drop of very dilute suspension on carbon-coated grids. The samples morphology and particle sizes were measured by a transmission electron microscopy (TEM) at 300 kV using a FEI Tecnai G2 F30 microscope.

23. SI line 99: XPS needs more information: e.g. charge neutralization (and method)? Sensitivity factor? ...

Reply: Thanks for the Reviewer's suggestion. We have revised the XPS part. The binding energy (BE) was corrected for surface charging by taking the C 1s peak of contaminant carbon as a reference at 284.58eV. Wagner – Database of Experimental Sensitivity Factors be used, based on $F1s = 1$.

24. SI line 105: ICP calibration method used?

Reply: Thanks for the Reviewer's suggestion. Inductively coupled plasma atomic emission spectroscopy (ICP-AES): The content of each element in the catalyst sample was determined by a PerkinElmer Optima 8300 inductively coupled plasma atomic emission spectrometer. The test process was as follows, Firstly, the catalyst sample was dissolved in hydrofluoric acid to ensure that it was completely dissolved and in a clear state. Finally, the solution was diluted to a suitable test range. The five standard solutions were prepared to construct the external standard curve. The content of elements in the samples was determined by external standard curve. We have

added this part into the revised manuscript.

25. Figure S2: Please add the initial HT sample result as well for comparison.

Reply: Thanks for the Reviewer's suggestion. We have added the initial HT sample result in the revised version.

Supplementary Fig. 2 XRD patterns of Cu/SiO₂ (reduced) prepared by hydrothermal method (HT), deposition–precipitation with ammonia (DPA), deposition–precipitation with urea (DPU), and an impregnation method (IM).

Reviewer #2:

Comments:

PET contributes significantly to plastic waste generation. In this work, Zhao and co-workers developed a new H₂ free method using Cu based catalyst to convert PET into xylene and ethylene glycol. The conversion efficiency is high and the reaction pathway has been well studied. The catalyst structure-activity correlation has also been convincingly described. Overall it is a nice piece of study that deserves to be published in Nat. Commun. after proper revision.

1) The stability of the catalyst. I feel this is a major limitation of the work. There is no adequate information on the stability of the Cu catalyst. Detailed characterizations of the spent Cu catalyst may be provided. Further, the reusability of the catalyst may be studied in more detail. If direct reuse is not possible, what is the reason for catalyst deactivation and whether it is possible to recover catalyst activity by certain treatment.

Reply: Thanks for the Reviewer's suggestion. After we tested the catalyst for four bathes, the catalyst deactivated obviously (Supplementary Table 9). According to XRD results, a large amount of Cu⁰ was reduced and the size of particles was increased (Supplementary Fig. 16a). In line with XRD results, TEM image showed that the Cu particles size was also increased after recycling tests (Supplementary Fig. 16b-c). XAES analysis proved that the ratio of Cu⁺/Cu⁰ drastically decreased from 1.80 to 0.57 (Supplementary Fig. 16f), which hindered the synergistic effect of Cu⁰ and Cu⁺ in the catalytic process. In addition, Cu²⁺ was partially reduced by the excess hydrogen produced by methanol dehydrogenation, as shown in XPS of Supplementary Fig. 16d-e.

Concerning on the recovery of catalyst activity, we tried to calcinate the used catalyst at 450°C in an air atmosphere for 4 h, and then reduced it at 450 °C in a hydrogen atmosphere for 4 h. The obtained catalyst continued for cycle testing and the results showed that PX yield attained 74.8%, 62.5% in the runs 1 and 2, while such recovered catalyst totally deactivated in the third run, probably due to the difficulty in rebalancing the ratio of Cu²⁺, Cu⁺ and Cu⁰ species in copper silicate catalysts.

Supplementary Table 9. Recycling tests of the conversion of PET over CuNa/SiO₂.

Run times	1	2	3	4
PX yield (%)	100	96.4	52.7	0

Reaction conditions: PET, 0.12 g; catalyst, 0.1 g; methanol, 30 mL; 210 °C; 6 h.

Supplementary Fig. 16 (a) XRD patterns of used CuNa/SiO₂ catalyst and CuNa/SiO₂ (reduced); TEM image of (b) used CuNa/SiO₂ catalyst and (c) CuNa/SiO₂ (reduced); XPS Cu 2p spectra of (d) used CuNa/SiO₂ catalyst and (e) CuNa/SiO₂ (reduced); Cu LMM XAES spectra of used CuNa/SiO₂ catalyst and CuNa/SiO₂ (reduced).

2) The authors may also comment on the applicability of the catalytic system beyond PET. If it is only applicable to PET, then sorting strategies have to be applied.

Reply: Thanks for the Reviewer's suggestion. Polybutylene terephthalate (PBT), which is used extensively in construction, electronics, and automotive parts, may bring

some plastic sediments through the construction industry and car rental industry on island. It has a great similarity to PET in structure which formed by polycondensation after esterification of dimethyl terephthalate (DMT) and 1,4-butanediol (BG). So we tested PBT under the same system and the gained results are very similar to PET conversion. At 210 °C, 100% yields of *p*-xylene and 1,4-butanediol were obtained from PBT in methanol, releasing 2.8 MPa gases (60% H₂). At the lower temperature of 190-200 °C, the yields of *p*-xylene were lowered to 95.7% and 60.6%, accompanied with the incremental pressure of 2.2 and 1.5 MPa. When the temperatures were decreased to 170 and 180 °C, only the monomer (DMT and EG) products were formed at the yields of 98.6% and 72.4%. DMT was not further converted due to the low gas pressures (0.5 and 0 MPa) at these two temperatures.

Considering that lots of plastic wastes on the island sediments landfill are mixture together, in this context, we used CuNa/SiO₂ to catalyze the mixture of PET and PBT at 210 °C, and the results showed that mixed plastics can be completely converted to *p*-xylene as well.

Supplementary Table 11. PBT conversion in methanol over CuNa/SiO₂ at different temperatures.

Temperature (°C)	PBT Conv. (%)	DMT yield (%)	PX yield (%)	By-product yield (%)		Incremental pressure at RT (MPa)	Gas composition (%)			
				methyl 4-methylbenzoate	4-methyl-benzyl alcohol		H ₂	CO	CO ₂	CH ₄
210	100	0	100	0	0	2.8	60	36	-	4
200	100	0	95.7	2.2	2.1	2.2	62	35	-	5
190	100	0	60.6	23.6	15.8	1.5	60	36	-	4
180	100	98.6	0	1.4	0	0.5	55	39	1	5
170	72.4	72.4	0	0	0	0	-	-	-	-

Reaction conditions: 0.12 g PBT, 0.1 g CuNa/SiO₂, 30 mL methanol, 6 h.

3) Methanol is used as solvent and hydrogen donor. How much methanol is decomposed during the process? Does the consumption of methanol match that of PET conversion?

Reply: Thanks for the Reviewer's suggestion. By calculating the consumption of methanol and the production of hydrogen, the experimental results showed that the values basically match. The following is the calculation process:

Reactions involved in the PET conversion process:

Real methanol consumption (according to liquid phase):

Experimental Method: 30 mL methanol was diluted 10 times in ethyl acetate with 0.1 mL tetrahydronaphthalene as the internal standard before the reaction and determined the response factor (f) by GC-MS using following formula.

$$f = \frac{A_S}{A_i} * \frac{V_i}{V_S}$$

The reacted solution was diluted 10 times with ethyl acetate, the residual amount of methanol is calculated by the following formula subsequently. According to the reduction of peak area, the residual methanol content was 97.6%.

$$V_{iu} = f * \frac{A_{iu}}{A_S} * V_S$$

$$n_{\text{consumption}}(\text{CH}_3\text{OH}) = 2.4\% * n(\text{CH}_3\text{OH}_{\text{original}}) = 0.0203 \text{ mol}$$

Therefore, the consumption of methanol is 0.0203 mol.

Real methanol consumption (according to gaseous phase):

The composition of the gaseous phase after reaction:

H₂ (60%), CO (36%), CH₄ (4%),

298 K, 3.4 MPa, 35 mL volume of reactor

Carbon Balance:

$$\begin{aligned} n_{\text{consumption}}(\text{CH}_3\text{OH}) &= n(\text{C}_{\text{gaseous phase}}) \\ &= n(\text{CO}) + n(\text{CH}_4) \\ &= 0.0192\text{mol} \end{aligned}$$

$$\begin{aligned} n_{\text{consume}}(\text{H}_2) &= 6n(\text{PET}) \\ &= 0.0038\text{mol} \end{aligned}$$

Hydrogen Balance:

$$\begin{aligned} n_{\text{production}}(\text{H}_2) &= n_{\text{residual amount}}(\text{H}_2) + n_{\text{consume}}(\text{H}_2) \\ &= n(\text{H}_2) + 3n(\text{CH}_4) + 6n(\text{PET}) \\ &= 0.0383\text{mol} \approx 2n_{\text{consumption}}(\text{CH}_3\text{OH}) \end{aligned}$$

Based on the above results, it can be concluded that the methanol consumption calculated from the product well matches the actual consumption of methanol and the generation of hydrogen.

4) It would also be good if the authors comment on how to purify products and reuse unreacted methanol.

Reply: Thanks for the Reviewer's suggestion. Both the products and methanol can be separated by simple distillation based on their different boiling points (see Supplementary Table 10 below).

Supplementary Table 10. Boiling points of different substances.

Substance	Boiling point (°C)
Methanol	64.7
p -Xylene	138.5
Ethylene Glycol	197.4

5) It is interesting to see the example of Phuket Island. Did the authors really use samples collected there, or it is just based on literature report? This needs to be made clear. If the sample from Phuket Island is not used, then it should be removed from Figure 5. Related parts in MS should be modified as well.

Reply: Thanks for the Reviewer's suggestion. A recent survey of beach sediment along the coastline of the Phuket Island showed that PET (mainly containing beverage bottles, plastic films, and microwave packaging) accounted for ca. 33.1% of the overall plastic sediment (Supplementary Fig. 17) ²⁹. Several common PET plastics that are available on the tourist island were chosen to convert, such as Coca-Cola bottles, McDonald's drink caps, disposable lunch boxes, packaging bags and even some

polyester clothes (Supplementary Fig. 18a). After simple treatment with the raw materials by scissors (Supplementary Fig. 18b), we obtained 100% yield of *p*-xylene from different sources of PET plastics at the same catalytic conditions. Considering that most plastic wastes on the island sediments landfill are mixture together, in this context, we used CuNa/SiO₂ to catalyze the mixture of PET and PBT at 210 °C, and the results showed that mixed plastics can be completely converted to *p*-xylene as well.

Table R2. Conversion of different PET plastic available on Phuket Island.

Substrate	PX yield (%)	By-product yield (%)	
		Methyl 4-methylbenzoate	4-methylbenzyl alcohol
Disposable lunch boxes	100	0	0
Coca-Cola bottles	100	0	0
Packaging bags	100	0	0
Polyester clothes	100	0	0
McDonald's drink caps	100	0	0

Reaction conditions: 0.12 g substrate, 0.1 g CuNa/SiO₂, 30 mL methanol, 6 h.

Supplementary Fig. 18 (a) Disposable lunch boxes, Coca-Cola bottles, packaging bags, polyester clothes and McDonald's drink caps; (b) Fragments of the plastics after processing; (c) The snapshots of the reaction system.

Reviewer #3:

Comments:

The manuscript “Converting waste PET plastics into automobile fuel and antifreeze components” by Zhao and coworkers describes a novel methodology for the depolymerization of PET waste using methanol as the solvent and hydrogen source, and a Cu-based catalyst. This new method is very important due to the use of an alcohol as reducing agent and also a non-toxic and earth abundant metal catalyst.

The work is well written and the discussion of the results appropriate, involving an extensive study of reaction conditions. Some aspects of the reaction mechanism were also included in this study. I recommend the publication of this manuscript in nature communications, after major revisions:

1. Authors should test this method using other alcohols, for example ethanol and isopropanol, as the hydrogen source and verify that PET depolymerization and *p*-xylene formation also occur.

Reply: Thanks for the Reviewer’s suggestion. We have supplemented the experiments on the PET conversion in ethanol and isopropanol in Supplementary Table 3. Experimental data showed that PET can be well alcoholized in both ethanol and isopropanol, obtain 80.3% and 73.5% yields of monomers after 0.5 h, respectively. However, the hydrogen released from ethanol and isopropanol decomposition over CuNa/SiO₂ was not sufficient, which attained only 0.7 and 0.8 MPa incremental pressure at ambient temperature, respectively. In comparison, methanol released as high as 3.8 MPa gases at identical conditions. In the further PET alcoholysis and hydrodeoxygenation tests in ethanol and isopropanol, the gained DMT monomers from PET were not hydrogenated and no *p*-xylene was formed in ethanol or isopropanol (Supplementary Table 3), probably due to the lack of sufficient released hydrogen in this catalytic system. This part has been supplemented into the new manuscript.

Supplementary Table 3. PET conversion in different solvents of methanol, ethanol and isopropanol.

Substrate	Catalyst	Solvent	Time (h)	PET Conv. (%)	Monomer yield (%)	PX yield (%)	Incremental pressure at RT (MPa)
PET	-	Methanol	0.5	100	100	0	0
-	CuNa/SiO ₂	Methanol	6	-	-	-	3.7
PET	CuNa/SiO ₂	Methanol	6	100	100	100	3.4
PET	-	Ethanol	0.5	80.3	80.3	0	0
-	CuNa/SiO ₂	Ethanol	6	-	-	-	0.7
PET	CuNa/SiO ₂	Ethanol	6	97.3	97.3	0	0.7
PET	-	Isopropanol	0.5	73.5	73.5	0	0
-	CuNa/SiO ₂	Isopropanol	6	-	-	-	0.8
PET	CuNa/SiO ₂	Isopropanol	6	87.1	87.1	0	0.8

Reaction conditions: 0.12 g PET, 0.1 g CuNa/SiO₂, 30 mL solvent, 210 °C.

2. Authors should also indicate whether the PET alcoholysis reaction can be carried out at temperatures below 210 °C and what yields are obtained.

Reply: Thanks for the Reviewer's suggestion. Experiments on the alcoholysis of PET at 170-210 °C in methanol for 30 min were showed in Supplementary Table 1.

At lower temperature of 170 °C , PET was depolymerized into dimethyl terephthalate (DMT) and ethylene glycol (EG) with yields of 63.3%, while at the higher temperature of 200 °C, PET was completely depolymerized to DMT and EG in methanol.

In addition, experiments on PET conversion over CuNa/SiO₂ in methanol at different temperatures were also supplemented (Supplementary Table 2). PET can be quantitatively alcoholized to DMT and EG at 180 °C in 6 h, while it achieved 63.9% monomer yields of DMT/EG at 170 °C. But when the temperature reached 200 °C, PX yield was increased to 93.6%.

Supplementary Table 1. PET alcoholysis at different temperatures in methanol.

Temperature (°C)	PET Conv. (%)	DMT yield (%)
210	100	100
200	100	100
190	78.6	78.6
180	63.3	63.3
170	19.8	19.8

Reaction conditions: 0.12 g PET, 30 mL methanol, 30min.

Supplementary Table 2. PET conversion over CuNa/SiO₂ catalysts at different temperatures in methanol.

Temperature (°C)	PET Conv. (%)	DMT yield (%)	PX yield (%)	By-product yield (%)	
				Methyl 4- methylbenzoate	4-methylbenzyl alcohol
210	100	0	100	0	0
200	100	0	93.6	6.1	0.3
190	100	0	53.8	32.4	13.8
180	100	99.4	0	0.6	0
170	63.9	63.9	0	0	0

Reaction conditions: 0.12 g PBT, 0.1 g CuNa/SiO₂, 30 mL methanol, 6 h.

3. To study the applicability of this method, it would be very interesting if the authors tested this method from the second most used polyester, polybutylene terephthalate (PBT), and checked if it is also possible to obtain *p*-xylene with good yields.

Reply: Thanks for the Reviewer's suggestion. We tested PBT conversion in methanol over CuNa/SiO₂ at different temperatures in the same catalytic system, and the results are very similar to PET conversion (Supplementary Table 11). At 210 °C, 100% yields of *p*-xylene and 1,4-butanediol were obtained from PBT in methanol, releasing 2.8 MPa gases (60% H₂). At the lower temperature of 190-200 °C, the yields of *p*-xylene were lowered to 95.7% and 60.6%, accompanied with the incremental

pressure of 2.2 and 1.5 MPa. When the temperatures were decreased to 170 and 180 °C, only the monomer (DMT and EG) products were formed at the yields of 98.6% and 72.4%. DMT was not further converted due to the low gas pressures (0.5 and 0 MPa) at these two temperatures.

Supplementary Table 11. PBT conversion in methanol over CuNa/SiO₂ at different temperatures.

Temperature (°C)	PBT Conv. (%)	DMT yield (%)	PX yield (%)	By-product yield (%)		Incremental pressure at RT (MPa)	Gas composition (%)			
				Methyl 4- methylbenzoate	4-Methylbenzyl alcohol		H ₂	CO	CO ₂	CH ₄
210	100	0	100	0	0	2.8	60	36	-	4
200	100	0	95.7	2.2	2.1	2.2	62	35	-	5
190	100	0	60.6	23.6	15.8	1.5	60	36	-	4
180	100	98.6	0	1.4	0	0.5	55	39	1	5
170	72.4	72.4	0	0	0	0	-	-	-	-

Reaction conditions: 0.12 g PBT, 0.1 g CuNa/SiO₂, 30 mL methanol, 6

Reviewers' Comments:

Reviewer #1:

Remarks to the Author:

The authors have addressed most of the technical questions sufficiently. Energetic/materials use calculations and comparisons to other work should however be included in the bulk of the paper as this is critical to the value of the work and puts it into perspective with other work.

I still have some concerns about the rationale though. I follow the thinking that on small islands there may not be the facilities to carry out chemical recycling and that waste might be too contaminated for physical recycling, but if chemical recycling is not possible, I highly doubt this conversion to fuel will be simpler. You will need just as much chemical facility to carry out this process in a safe manner and carry out the separations that are needed to run the process optimally. I would think a direct pyrolysis to carbon compounds/liquid fuel would be more efficient and take less in terms of industrial facility. Burning PET directly for electricity generation would also be more efficient and not require all these extra steps...

Reviewer #2:

Remarks to the Author:

The authors have well addressed my initial comments. Acceptance in the revised form is therefore recommended.

Reviewer #3:

Remarks to the Author:

Authors responded exhaustively to issues raised in the first round.

Reviewer #1:

The authors have addressed most of the technical questions sufficiently. Energetic/materials use calculations and comparisons to other work should however be included in the bulk of the paper as this is critical to the value of the work and puts it into perspective with other work.

I still have some concerns about the rationale though. I follow the thinking that on small islands there may not be the facilities to carry out chemical recycling and that waste might be too contaminated for physical recycling, but if chemical recycling is not possible, I highly doubt this conversion to fuel will be simpler. You will need just as much chemical facility to carry out this process in a safe manner and carry out the separations that are needed to run the process optimally. I would think a direct pyrolysis to carbon compounds/liquid fuel would be more efficient and take less in terms of industrial facility. Burning PET directly for electricity generation would also be more efficient and not require all these extra steps...

Reply: Thanks for the Reviewer's suggestion. In the bulk text and supporting information of this manuscript, we have added the comparison of energetic/materials calculation for PET conversion (Table R1) ¹⁻³.

We investigated the data on thermal pyrolysis and incineration of PET. In the absence of the catalyst, a high gas product yield (76.9 wt%) as well as a liquid oil yield of 23.1 wt% was obtained from PET pyrolysis. And the pyrolysis oil from PET contained 49.93% benzoic acid, making the oil products unfavorable for qualified fuels due to its corrosive property ⁴. Furthermore, benzoic acid clogs stainless steel pipelines and heat exchangers, causing serious problems when running on an industrial scale ^{5,6}.

Incineration is considered as an inefficient method for energy recycling, and pays high environmental costs (especially for islands) (Figure R1) ⁷. Aryan et al. calculated the amount for net electricity generation from incineration of 1 ton of PET waste using Dulong's equation. The PET waste heating value was 22381.92 kJ/kg. Considering the conversion efficiency (32%) and the loss of energy (10%), electricity generated is calculated to be 1253kWh of every ton of PET waste in the thermal

power plant, less than half of every ton of PE plastic (2760 kWh)⁸. Moreover, emissions and sources required related are listed as blow (Table R2), which showed that a lots of pollutant gases were formed after incineration of PET.

In comparison, based on our developed new process using methanol as solvent and hydrogen source, PET can be converted to 100% yield of *p*-xylene and ethylene glycol using low-cost Cu-based catalysts in one-pot process. The obtained products can be used as vehicle fuels and antifreeze components after simple separation. No pollutant gases are produced. And local digestion of the products from PET conversion does not require long-distance transportation, which makes this process to be quite economic. Based on these considerations, we believe the new catalytic system in this work is more competitive.

Table R1. Summary of main results in literature for PET conversion.

Catalyst	Noble metal	T (°C)	Reaction time (min)	Yield arene (%)	Solvent/PET mass ratio	Catalyst/PET mass ratio	Energy economy (ε) (°C ⁻¹ *min ⁻¹)	Environmental factor (a.u)	Environmental energy impact (ξ) (°C*min)	Ref.
Ru/Nb ₂ O ₅	Yes	200	720	87.1%	100.0	1.00	6.050E-6	23.74	3923967	2
Ru/Nb ₂ O ₅	Yes	220	720	90.4%	75.0	1.00	5.710E-6	19.70	3450088	3
CuNa/SiO ₂	No	210	360	100%	197.5	0.83	1.323E-5	37.19	2811035	This work
CuNa/SiO ₂	No	210	360	100%	98.75	0.83	1.323E-5	19.40	1466364	This work
CuNa/SiO ₂	No	210	360	100%	65.83	0.83	1.323E-5	13.41	1013605	This work

Figure R1. The Pyramid of Plastic Waste Management. (Source: BCG, 2019)

Table R2. Emissions due to total electricity generation (amount equals to electricity generated by incineration 1 tonne of PET waste) from coal-based thermal power plant. (Source: Aryan et al., 2019)

S.N.	Emissions kg/tonne for production of 1253 kWh of electricity	
1	CO ₂	1141.48kg
2	CO	0.201kg
3	SO ₂	10.15kg
4	NO	2.62kg
Resource required		
5	Coal required	877.1kg

Ref.

- [1] Barnard, E., Rubio Arias, J. J. & Thielemans, W. Chemolytic depolymerisation of PET: a review. *Green Chemistry* 23, 3765-3789 (2021).
- [2] Jing, Y. et al. Towards the Circular Economy: Converting aromatic plastic waste back to arenes over a Ru/Nb₂O₅ catalyst. *Angewandte Chemie International Edition* 60, 5527-5535 (2021).
- [3] Lu, S. et al. H₂-free plastic conversion: converting PET back to BTX by unlocking hidden hydrogen. *ChemSusChem* 14, 4242-4250 (2021).
- [4] Peng, Y. et al. A review on catalytic pyrolysis of plastic wastes to high-value products. *Energy Conversion and Management* 254, 115243 (2022).
- [5] Lee, J., Lee, T., Tsang, Y. F., Oh, J.-I. & Kwon, E. E. Enhanced energy recovery from polyethylene terephthalate via pyrolysis in CO₂ atmosphere while suppressing acidic chemical species. *Energy Conversion and Management* 148, 456-460 (2017).
- [6] Veksha, A. et al. Technical and environmental assessment of laboratory scale approach for sustainable management of marine plastic litter. *Journal of Hazardous Materials* 421, 126717 (2022).
- [7] Boston Consulting Group. A circular solution to plastic waste (2019).
- [8] Aryan, Y., Yadav, P. & Samadder, S. R. Life Cycle assessment of the existing and proposed plastic waste management options in India: A case study. *Journal of Cleaner Production* 211, 1268-1283 (2019).

Reviewer #2:

Comments:

The authors have well addressed my initial comments. Acceptance in the revised form is therefore recommended.

Reply: We are grateful to the reviewer's positive comment.

Reviewer #3:

Comments:

Authors responded exhaustively to issues raised in the first round.

Reply: We are grateful to the reviewer's positive comment.

We hope the revised manuscript is now suitable to be published in Nature Communications. If you require any further information, please do not hesitate to contact me.